# Stochastic yield catastrophes and robustness in self-assembly

**Florian M Gartner†, Isabella R Graf†, Patrick Wilke†, Philipp M Geiger, Erwin Frey\***

Arnold Sommerfeld Center for Theoretical Physics (ASC) and Center for NanoScience (CeNS), Department of Physics, Ludwig-Maximilians-Universität München, München, Germany

**Abstract** A guiding principle in self-assembly is that, for high production yield, nucleation of structures must be significantly slower than their growth. However, details of the mechanism that impedes nucleation are broadly considered irrelevant. Here, we analyze self-assembly into finite-sized target structures employing mathematical modeling. We investigate two key scenarios to delay nucleation: (i) by introducing a slow activation step for the assembling constituents and, (ii) by decreasing the dimerization rate. These scenarios have widely different characteristics. While the dimerization scenario exhibits robust behavior, the activation scenario is highly sensitive to demographic fluctuations. These demographic fluctuations ultimately disfavor growth compared to nucleation and can suppress yield completely. The occurrence of this stochastic yield catastrophe does not depend on model details but is generic as soon as number fluctuations between constituents are taken into account. On a broader perspective, our results reveal that stochasticity is an important limiting factor for self-assembly and that the specific implementation of the nucleation process plays a significant role in determining the yield.

**\*For correspondence:**
frey@lmu.de

†These authors contributed equally to this work

**Competing interests:** The authors declare that no competing interests exist.

## Introduction

Efficient and accurate assembly of macromolecular structures is vital for living organisms. Not only must resource use be carefully controlled, but malfunctioning aggregates can also pose a substantial threat to the organism itself (*Jucker and Walker, 2013*; *Drummond and Wilke, 2009*). Furthermore, artificial self-assembly processes have important applications in a variety of research areas like nanotechnology, biology, and medicine (*Zhang, 2003*; *Whitesides and Grzybowski, 2002*; *Whitesides et al., 1991*). In these areas, we find a broad range of assembly schemes. For example, while a large number of viruses assemble capsids from identical protein subunits, some others, like the Escherichia virus T4, form highly complex and heterogeneous virions encompassing many different types of constituents (*Zlotnick et al., 1999*; *Zlotnick, 2003*; *Hagan, 2014*; *Leiman et al., 2010*). Furthermore, artificially built DNA structures can reach up to Gigadalton sizes and can, in principle, comprise an unlimited number of different subunits (*Ke et al., 2012*; *Reinhardt and Frenkel, 2014*; *Gerling et al., 2015*; *Wagenbauer et al., 2017*). Notwithstanding these differences, a generic self-assembly process always includes three key steps: First, subunits must be made available, for example by gene expression, or rendered competent for binding, for example by nucleotide exchange (*Alberts and Johnson, 2015*; *Chen et al., 2008*; *Whitelam, 2015*) ('activation'). Second, the formation of a structure must be initiated by a nucleation event ('nucleation'). Due to cooperative or allosteric effects in binding, there might be a significant nucleation barrier (*Chen et al., 2008*; *Jacobs and Frenkel, 2015*; *Sear, 2007*; *Lazaro and Hagan, 2016*; *Hagan and Elrad, 2010*). Third, following nucleation, structures grow via aggregation of substructures ('growth'). To avoid kinetic traps that may occur due to irreversibility or very slow disassembly of substructures (*Hagan et al., 2011*; *Grant et al., 2011*), structure nucleation must be significantly slower than growth

**eLife digest** The self-assembly of a large biological molecule from small building blocks is like finishing a puzzle of magnetic pieces by shaking the box. Even though each piece of the puzzle is attracted to its correct neighbours, the limited control makes it very hard to finish the puzzle in a short amount of time.

The problem becomes even more difficult if several copies of the same puzzle are assembled in one box. If several puzzles start at the same time, the different parts might steal pieces from each other, making it impossible to successfully complete any of the puzzles. This is called a depletion trap. If the box is only shaken and there is no real control over individual pieces, these traps occur at random.

Overcoming these random depletion traps is an important challenge when assembling nanostructures and other artificial molecules designed by humans without wasting many, potentially expensive, components. Previous studies have shown that when multiple copies of the same structure are assembled simultaneously, slowing the rate of initiation increases the yield of correctly-made structures. This prevents new structures from stealing pieces from existing structures before they are fully completed.

Now, Gartner, Graf, Wilke et al. have used a mathematical model to show that changing the way initiation is delayed leads to different yields. This was especially true for small systems where fluctuations in the availability of the different pieces strongly enhanced the initiation of new structures. In these cases, the self-assembly process terminated undesirably with many incomplete structures.

Nanostructures have various applications ranging from drug delivery to robotics. These findings suggest that in order to efficiently assemble biological molecules, the concentrations of the different building blocks need to be tightly controlled. A question for further research is to investigate strategies that reduce fluctuations in the availability of the building blocks to develop more efficient assembly protocols.

(*Zlotnick et al., 1999*; *Ke et al., 2012*; *Reinhardt and Frenkel, 2014*; *Wei et al., 2012*; *Jacobs et al., 2015*; *Hagan and Elrad, 2010*). Physically speaking, there are no irreversible reactions. However, in the biological context, self-assembly describes the (relatively fast) formation of long-lasting, stable structures. Therefore, at least part of the assembly reactions are often considered to be irreversible on the time scale of the assembly process. In this manuscript we investigate, for a given target structure, whether the nature of the specific mechanism employed in order to slow down nucleation influences the yield of assembled product. To address this question, we examine a generic model that incorporates the key elements of self-assembly outlined above.

## Model definition

We model the assembly of a fixed number of well-defined target structures from limited resources. Specifically, we consider a set of $S$ different species of constituents denoted by $1, \ldots, S$ which assemble into rings of size $L$. The cases $S = 1$ and $1 < S \leq L$ ($S = L$) are denoted as homogeneous and partially (fully) heterogeneous, respectively. The homogeneous model builds on previous work on virus capsid (*Chen et al., 2008*; *Hagan et al., 2011*), linear protein filament assembly (*Michaels et al., 2016*; *Michaels et al., 2017*; *D'Orsogna et al., 2012*) and aggregation and polymerization models (*Krapivsky et al., 2010*). The heterogeneous model in turn links to previous model systems used to study, for example, DNA-brick-based assembly of heterogeneous structures (*Murugan et al., 2015*; *Hedges et al., 2014*; *D'Orsogna et al., 2013*). We emphasize that, even though strikingly similar experimental realizations of our model exist (*Gerling et al., 2015*; *Wagenbauer et al., 2017*; *Praetorius and Dietz, 2017*), it is not intended to describe any particular system. The ring structure represents a general linear assembly process involving building blocks with equivalent binding properties and resulting in a target of finite size. The main assumption in the ring model is that the different constituents assemble linearly in a sequential order. In many biological self-assembling systems like bacterial flagellum assembly or biogenesis of the ribosome subunits the assumption of a linear binding sequence appears to be justified (*Peña et al., 2017*; *Chevance and Hughes, 2008*). In order

to test the validity of our results beyond these constraints we also perform stochastic simulations of generalized self-assembling systems that do not obey a sequential binding order: i) by explicitly allowing for polymer-polymer bindings and ii) by considering the assembly of finite sized squares that grow independently in two dimensions (see Figures 6 and 7).

The assembly process starts with $N$ inactive monomers of each species. We use $C = N/V$ to denote the initial concentration of each monomer species, where $V$ is the reaction volume. Monomers are activated independently at the same per capita rate $\alpha$, and, once active, are available for binding. Binding takes place only between constituents of species with periodically consecutive indices, for example 1 and 2 or $S$ and 1 (leading to structures such as . . .1231. . . for $S = 3$); see *Figure 1*. To avoid ambiguity, we restrict ring sizes to integer multiples of the number of species $S$. Furthermore, we neglect the possibility of incorrect binding, for example species 1 binding to 3 or $S-1$. Polymers, that is incomplete ring structures, grow via consecutive attachment of monomers. For simplicity, polymer-polymer binding is disregarded at first, as it is typically assumed to be of minor importance (*Zlotnick et al., 1999*; *Chen et al., 2008*; *Murugan et al., 2015*; *Haxton and Whitelam, 2013*). To probe the robustness of the model, later we consider an extended model including polymer-polymer binding for which the results are qualitatively the same (see Figure 6 and the discussion). Furthermore, it has been observed that nucleation phenomena play a critical role for self-assembly processes (*Ke et al., 2012*; *Wei et al., 2012*; *Reinhardt and Frenkel, 2014*; *Chen et al., 2008*). So it is in general necessary to take into account a critical nucleation size, which marks the transition between slow particle nucleation and the faster subsequent structure growth (*Michaels et al., 2016*; *Lazaro and Hagan, 2016*; *Morozov et al., 2009*; *Murugan et al., 2015*). We denote this critical nucleation size by $L_{\mathrm{nuc}}$, which in terms of classical nucleation theory corresponds to the structure size at which the free energy barrier has its maximum. For $l < L_{\mathrm{nuc}}$ attachment of monomers to existing structures and decay of structures (reversible binding) into monomers take place at size-dependent reaction rates $\mu_l$ and $\delta_l$, respectively (*Figure 1*). Here, we focus on identical rates $\mu_l = \mu$ and $\delta_l = \delta$. A discussion of the general case is given in Appendix 4. Above the nucleation size, polymers grow by attachment of monomers with reaction rate $\nu \geq \mu$ per binding site. As

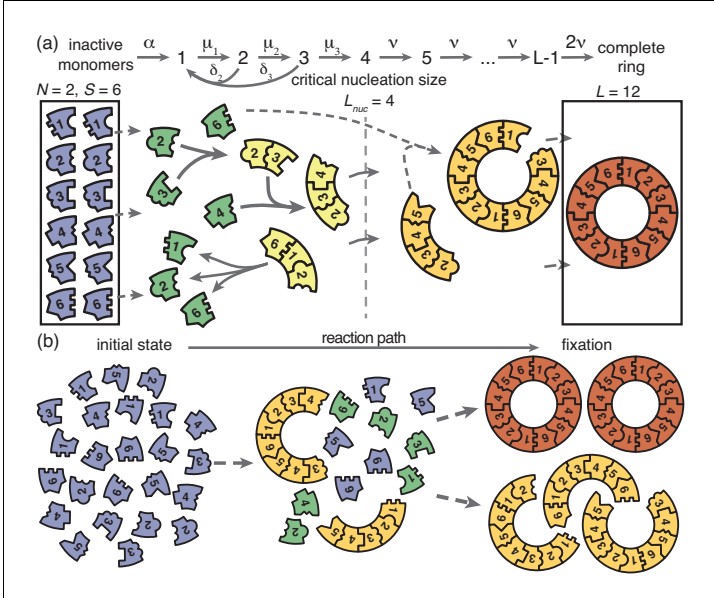

**Figure 1.** Schematic description of the model. (**a**) Rings of size $L$ are assembled from $S$ different particle species. $N$ monomers of each species are initially in an inactive state (blue) and are activated at the same per-capita rate $\alpha$. Once active (green), species with periodically consecutive index can bind to each other. Structures grow by attachment of single monomers. Below a critical nucleation size ($L_{\mathrm{nuc}}$), structures of size $l$ (light yellow) grow and decay into monomers at size-dependent rates $\mu_l$ and $\delta_l$, respectively. Above the critical size, polymers (dark yellow) are stable and grow at size-independent rate $\nu$ until the ring is complete (the absorbing state; red). (**b**) Illustration of depletion traps. If nucleation is slow compared to growth, initiated structures are likely to be completed. Otherwise, many stable nuclei will form that cannot be completed before resources run out.

we consider successfully nucleated structures to be stable on the observational time scales, monomer detachment from structures above the critical nucleation size is neglected (irreversible binding) (*Murugan et al., 2015*; *Chen et al., 2008*). Complete rings neither grow nor decay (absorbing state).

We investigate two scenarios for the control of nucleation speed, first separately and then in combination. For the 'activation scenario' we set $\mu = \nu$ (all binding rates are equal) and control the assembly process by varying the activation rate $\alpha$. For the 'dimerization scenario' all particles are inherently active ($\alpha \to \infty$) and we control the assembly process by varying the dimerization rate $\mu$ (we focus on $L_{\mathrm{nuc}} = 2$). It has been demonstrated previously in *Chen et al. (2008)* and (*Endres and Zlotnick, 2002*; *Hagan and Elrad, 2010*; *Morozov et al., 2009*) that either a slow activation or a slow dimerization step are suitable in principle to retard nucleation and favour growth of the structures over the initiation of new ones. We quantify the quality of the assembly process in terms of the assembly yield, defined as the number of successfully assembled ring structures relative to the maximal possible number $NS/L$. Yield is measured when all resources have been used up and the system has reached its final state. We do not discuss the assembly time in this manuscript, however, in Appendix 5 we show typical trajectories for the time evolution of the yield in the activation and dimerization scenario. If the assembly product is stable (absorbing state), the yield can only increase with time. Consequently, the final yield constitutes the upper limit for the yield irrespective of additional time constraints. Therefore, the final yield is an informative and unambiguous observable to describe the efficiency of the assembly reaction.

We simulated our system both stochastically via Gillespie's algorithm (*Gillespie, 2007*) and deterministically as a set of ordinary differential equations corresponding to chemical rate equations (see Appendix 1).

## Results

### Deterministic behavior in the macroscopic limit

First, we consider the macroscopic limit, $N \gg 1$, and investigate how assembly yield depends on the activation rate $\alpha$ (activation scenario) and the dimerization rate $\mu$ (dimerization scenario) for $L_{\mathrm{nuc}} = 2$. Here, the deterministic description coincides with the stochastic simulations (*Figure 2a and b*). For both high activation and high dimerization rates, yield is very poor. Upon decreasing either the activation rate (*Figure 2a*) or the dimerization rate (*Figure 2b*), however, we find a threshold value, $\alpha_{\mathrm{th}}$ or $\mu_{\mathrm{th}}$, below which a rapid transition to the perfect yield of 1 is observed both in the deterministic and stochastic simulation. By exploiting the symmetries of the system with respect to relabeling of species, one can show that, in the deterministic limit, the behavior is independent of the number of species $S$ (for fixed $L$ and $N$, see Appendix 1). Consequently, all systems behave equivalently to the homogeneous system and yield becomes independent of $S$ in this limit. Note, however, that equivalent systems with differing $S$ have different total numbers of particles $SN$ and hence assemble different total numbers of rings.

Decreasing the activation rate reduces the concentration of active monomers in the system. Hence growth of the polymers is favored over nucleation, because growth depends linearly on the concentration of active monomers while nucleation shows a quadratic dependence. Likewise, lower dimerization rates slow down nucleation relative to growth. Both mechanisms therefore restrict the number of nucleation events, and ensure that initiated structures can be completed before resources become depleted (see *Figure 2c and d*).

Mathematically, the deterministic time evolution of the polymer size distribution $c(l, t)$ is described by an advection-diffusion equation (*Endres and Zlotnick, 2002*; *Yvinec et al., 2012*) with advection and diffusion coefficients depending on the instantaneous concentration of active monomers (see Appendix 2). Solving this equation results in the wavefront of the size distribution advancing from small to large polymer sizes (*Figure 2e*). Yield production sets in as soon as the distance travelled by this wavefront reaches the maximal ring size $L$. Exploiting this condition, we find that in the deterministic system for $L_{\mathrm{nuc}} = 2$, a non-zero yield is obtained if either the activation rate or the dimerization rate remains below a corresponding threshold value, that is if $\alpha < \alpha_{\mathrm{th}}$ or $\mu < \mu_{\mathrm{th}}$, where

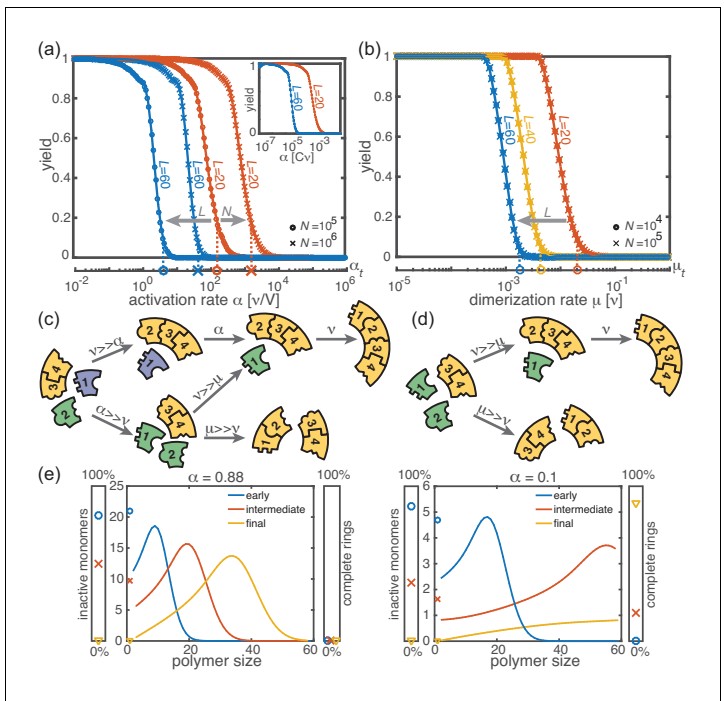

**Figure 2.** Deterministic behavior in the macroscopic limit $N \gg 1$. (a, b) Yield for different particle numbers $N$ (symbols) and ring sizes $L$ (colors) for $L_{\mathrm{nuc}} = 2$. Decreasing either (a) the activation rate ('activation scenario': $\mu = \nu$) or (b) the dimerization rate ('dimerization scenario': $\alpha \to \infty$) achieves perfect yield. The stochastic simulation results (symbols) average over 16 realizations and agree exactly with the integration of the chemical rate equations (lines). The threshold values (*Equation 1*) are indicated by the vertical dashed lines. Plotting yield against the dimensionless quantity $\alpha/(\nu C)$ causes the curves for different $C$ to collapse into a single master curve (inset in a). For both scenarios there is no dependency on the number of species $S$ in the deterministic limit. (c, d) Illustration showing how depletion traps are avoided by either slow activation (c) or slow dimerization (d). If the activation or the dimerization rate is small (large) compared to the growth rate, assembly paths leading to complete rings are favored (disfavored). The color scheme is the same as in *Figure 1*. (e) Deterministically, the size distribution of polymers behaves like a wave, and is shown for large and small activation rate for $L = 60$, $L_{\mathrm{nuc}} = 2$, $N = 10000$ and $\mu = \nu = 1$. The distributions are obtained from a numerical integration of the deterministic mean-field dynamics, *Equation 6*, and are plotted for early, intermediate and final simulation times. The respective percentage of inactive monomers and complete rings is indicated by the symbols in the scale bar on the left or right.

$$\alpha_{\mathrm{th}} = P_\alpha \frac{\nu}{\mu} \frac{\nu C}{(L - \sqrt{L})^3} \quad \text{and} \quad \mu_{\mathrm{th}} = P_\mu \frac{\nu}{(L - \sqrt{L})^2} \tag{1}$$

(see Appendix 3) with proportionality constants $P_\alpha = [\sqrt{\pi}\Gamma(2/3)/\Gamma(7/6)]^3/3 \approx 5.77$ and $P_\mu = \pi^2/2 \approx 4.93$. These relations generalize previous results (*Morozov et al., 2009*) to finite activation rates and for heterogeneous systems. A comparison between the threshold values given by *Equation 1* and the simulated yield curves is shown in *Figure 2a,b*. The relations highlight important differences between the two scenarios (where $\alpha \to \infty$ and $\mu = \nu$, respectively): While $\alpha_{\mathrm{th}}$ decreases cubically with the ring size $L$, $\mu_{\mathrm{th}}$ does so only quadratically. Furthermore, the threshold activation rate $\alpha_{\mathrm{th}}$ increases with the initial monomer concentration $C$. Consequently, for fixed activation rate, the yield can be optimized by increasing $C$. In contrast, the threshold dimerization rate is independent of $C$ and the yield curves coincide for $N \gg 1$. Finally, if $\alpha$ is finite and $\mu < \nu$, the interplay between the two slow-nucleation scenarios may lead to enhanced yield. This is reflected by the factor $\nu/\mu$ in $\alpha_{\mathrm{th}}$, and we will come back to this point later when we discuss the stochastic effects.

In summary, for large particle numbers ($N \gg 1$), perfect yield can be achieved in two different ways, independently of the heterogeneity of the system - by decreasing either the activation rate (activation scenario) or the dimerization rate (dimerization scenario) below its respective threshold value.

## Stochastic effects in the case of reduced resources

Next, we consider the limit where the particle number becomes relevant for the physics of the system. In the activation scenario, we find a markedly different phenomenology if resources are sparse. *Figure 3a* shows the dependence of the average yield on the activation rate for different, low particle numbers in the completely heterogeneous case ($S = L$). Here, we restrict our discussion to the average yield. The error of the mean is negligible due to the large number of simulations used to calculate the average yield. Still, due to the randomness in binding and activation, the yield can differ between simulations. A figure with the average yield and its standard deviation is shown in Appendix 6. For very low and very high average yield, the standard deviation has to be small due to the boundedness of the yield. For intermediate values of the average, the standard deviation is highest but still small compared to the average yield. Thus, the average yield is meaningful for the essential understanding of the assembly process. Whereas the deterministic theory predicts perfect yield for small activation rates, in the stochastic simulation yield saturates at an imperfect value $y_{max} < 1$. Reducing the particle number $N$ decreases this saturation value $y_{max}$ until no finished structures are produced ($y_{max} \to 0$). The magnitude of this effect strongly depends on the size of the target structure $L$ if the system is heterogeneous. *Figure 3c* shows a diagram characterizing different regimes for the saturation value of the yield, $y_{max}(N, L)$, in dependence of the particle number $N$ and the size of the target structure $L$ for fully heterogeneous systems ($S = L$). We find that the threshold particle number $N_y^{th}$ necessary to obtain a fixed yield $y$ increases nonlinearly with the target size $L$. For the depicted range of $L$, the dependence of the threshold for nonzero yield, $N_{>0}^{th}$, on $L$ can approximately be described by a power-law: $N_{>0}^{th} \sim L^\xi$, with exponent $\xi \approx 2.8$ for $L \leq 600$. Consequently, for

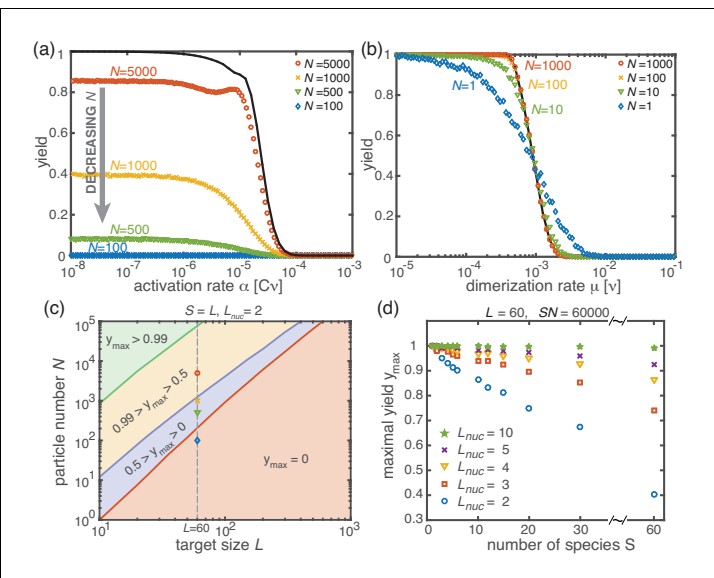

**Figure 3.** Stochastic effects in the case of reduced resources. (a, b) Yield of the fully heterogeneous system ($S = L$) for reduced number of particles (symbols) for $L = 60$ and $L_{nuc} = 2$ averaged over 1024 ensembles. In the activation scenario, at low activation rates the yield saturates at an imperfect value $y_{max}$, which decreases with the number of particles (a). This finding disagrees with the deterministic prediction (black line) of perfect yield for $\alpha \to 0$. In contrast, the dimerization scenario robustly exhibits the maximal yield of 1 for small $N$, in agreement with the deterministic prediction (black line) (b). (c) Diagram showing different regimes of $y_{max}(N, L)$ in dependence of the particle number $N$ and target size $L$ (for the fully heterogeneous system $S = L$) as obtained from stochastic simulations in the limit $\alpha \to 0$. The minimal number of particles necessary to obtain a fixed yield increases in a strongly nonlinear way with the target size. The symbols along the line $L = 60$ represent the saturation values of the yield curves in (a). (d) Dependence of $y_{max}$ on the number of species $S$ for fixed $L = 60$ and fixed number of ring structures $NS/L$. Symbols indicate different values of the critical nucleation size $L_{nuc}$. The impact of stochastic effects strongly depends on the number of species under the constraint of a fixed total number of particles $NS$ and fixed target size $L$. The homogeneous system is not subject to stochastic effects at all. Higher reversibility for larger $L_{nuc}$ also mitigates stochastic effects.

$L = 600$ already more than $10^5$ rings must be assembled in order to obtain a yield larger than zero. In Appendix 8 we included two additional plots that show the dependence of $y_\mathrm{max}$ on $N$ for fixed $L$ and the dependence on $L$ for fixed $N$, respectively. The suppression of the yield is caused by fluctuations (see explanation below) and is not captured by a deterministic description. Because these stochastic effects can decrease the yield from a perfect value in a deterministic description to zero (see **Figure 3a**), we term this effect 'stochastic yield catastrophe'. For fixed target size $L$ and fixed maximum number of target structures $\frac{NS}{L}$, $y_\mathrm{max}$ increases with decreasing number of species, see **Figure 3d**. In the fully homogeneous case, $S = 1$, a perfect yield of 1 is always achieved for $\alpha \to 0$. The decrease of the maximal yield with the number of species $S$ thus suggests that, in order to obtain high yield, it is beneficial to design structures with as few different species as possible. In large part this effect is due to the constraint $SN = \mathrm{const}$, whereby the more homogeneous systems (small $S$) require larger numbers of particles per species $N$ and, correspondingly, exhibit less stochasticity. If $N$ is fixed instead of $SN$, the yield still initially decreases with increasing number of species $S$ but then quickly reaches a stationary plateau and gets independent of $S$ for $S \gg 1$, see Appendix 7. Moreover, increasing the nucleation size $L_\mathrm{nuc}$, and with it the reversibility of binding, also increases $y_\mathrm{max}$, see **Figure 3(d)**. This indicates that, beside heterogeneity of the target structure, irreversibility of binding on the relevant time scale makes the system susceptible to stochastic effects.

The stochastic yield catastrophe is mainly attributable to fluctuations in the number of active monomers. In the deterministic (mean-field) equation the different particle species evolve in balanced stoichiometric concentrations. However, if activation is much slower than binding, the number of active monomers present at any given time is small, and the mean-field assumption of equal concentrations is violated due to fluctuations (for $S>1$). Activated monomers then might not fit any of the existing larger structures and would instead initiate new structures. **Figure 4a** illustrates this effect and shows how fluctuations in the availability of active particles lead to an enhanced nucleation and, correspondingly, to a decrease in yield. Due to the effective enhancement of the nucleation rate, the resulting polymer size distribution has a higher amplitude than that predicted deterministically (**Figure 4b**) and the system is prone to depletion traps. A similar broadening of the size distribution has been reported in the context of stochastic coagulation-fragmentation of identical particles (**D'Orsogna et al., 2015**).

In the dimerization scenario, in contrast, there is no stochastic activation step. All particles are available for binding from the outset. Consequently, stochastic effects do not play an essential role in the dimerization scenario and perfect yield can be reached robustly for all system sizes, regardless of the number of species $S$ (**Figure 3(b)**).

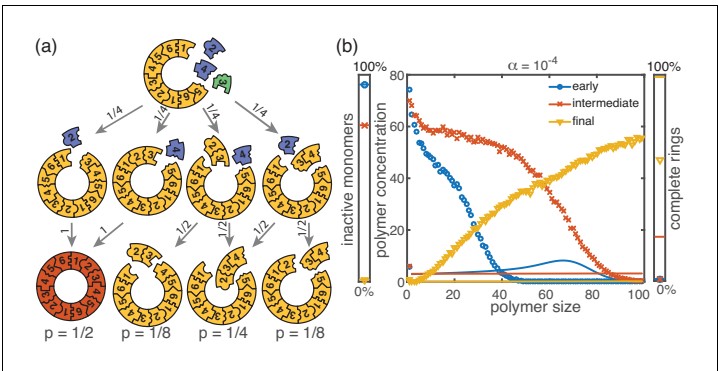

**Figure 4.** Cause and effect of stochasticity in the activation scenario. (**a**) Illustration of the significance of stochastic effects when resources are sparse. Arrows indicate possible transitions and the probabilities in the depicted situation for sufficiently small activation rate $\alpha$. For small $\alpha$, the random order of activation alone determines the availability of monomers and therefore the order of binding. In the depicted situation, the complete structure is assembled only with probability 1/2. In all other cases, only fragments of the structure are assembled such that the final yield is decreased. (**b**) Polymer size distribution for the activation scenario (symbols) as obtained from stochastic simulations, in comparison with its deterministic prediction (lines) for $S = L = 100$, $N = 1000$ and $L_\mathrm{nuc} = 2$. Due to the enhanced number of nucleation events, the stochastic wave encompasses far more structures and moves more slowly. As a result, it does not quite reach the absorbing boundary.

## Non-monotonic yield curves for a combination of slow dimerization and activation

So far, the two implementations of the 'slow nucleation principle' have been investigated separately. Surprisingly, we observe counter-intuitive behavior in a mixed scenario in which both dimerization and activation occur slowly (i.e., $\mu < \nu$, $\alpha < \infty$). *Figure 5* shows that, depending on the ratio $\mu/\nu$, the yield can become a non-monotonic function of $\alpha$. In the regime where $\alpha$ is large, nucleation is dimerization-limited; therefore activation is irrelevant and the system behaves as in the dimerization scenario for $\alpha \to \infty$. Upon decreasing $\alpha$ we then encounter a second regime, where activation and dimerization jointly limit nucleation. The yield increases due to synergism between slow dimerization and activation (see $\mu/\nu$ dependence of $\alpha_{\mathrm{th}}$, *Equation 1*), whilst the average number of active monomers is still high and fluctuations are negligible. Finally, a stochastic yield catastrophe occurs if $\alpha$ is further reduced and activation becomes the limiting step. This decline is caused by an increase in nucleation events due to relative fluctuations in the availability of the different species ('fluctuations between species'). This contrasts the deterministic description where nucleation is always slower for smaller activation rate. Depending on the ratio $\mu/\nu$, the ring size $L$ and the particle number $N$, maximal yield is obtained either in the dimerization-limited (red curves, *Figure 5*), activation-limited (blue curve, *Figure 5b*) or intermediate regime (green and orange curves, *Figure 5*).

## Robustness of the results to model modifications

In our model, the reason for the stochastic yield catastrophe is that - due to fluctuations between species - the effective nucleation rate is strongly enhanced. Hence, if binding to a larger structure is temporarily impossible, activated monomers tend to initiate new structures, causing an excess of

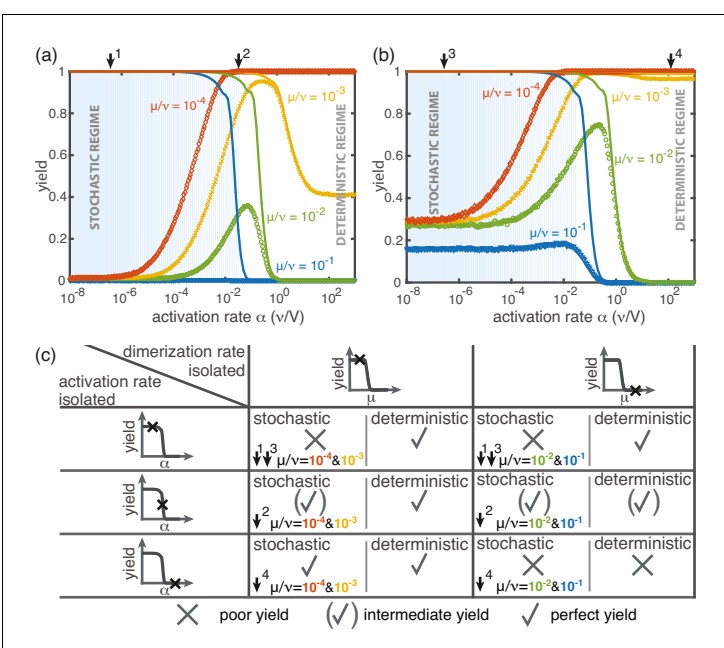

**Figure 5.** Yield for a combination of slow dimerization and activation. (a, b) Dependence of the yield of the fully heterogeneous system on the activation rate $\alpha$ for $N = 100$ and different values of the dimerization rate (colors/symbols) for $L = 60$ (a) and $L = 40$ (b) (averaged over 1024 ensembles). For large activation rates the yield behaves deterministically (lines). In contrast, for small activation rates, stochastic effects (blue shading) lead to a decrease in yield. Depending on the parameters, the yield maximum is attained in either the deterministic, stochastic or intermediate regime. (c) Table summarizing the qualitative behavior of the yield (poor/intermediate/perfect) for a combination of dimerization and activation rates for both the deterministic and the stochastic limit. The columns correspond to low and high values of the dimerization rate, as indicated by the marker in the corresponding deterministic yield curve at the top of the column. Similarly, the rows correspond to low, intermediate and high activation rates. Arrows and colors indicate where and for which curve this behavior can be observed in (a) and (b). Deviations between the deterministic and stochastic limits are most prominent for low activation rates.

structures that ultimately cannot be completed. Natural questions that arise are whether (i) relaxing the constraint that polymers cannot bind other polymers or (ii) abandoning the assumption of a linear assembly path, will resolve the stochastic yield catastrophe. To answer these questions, we performed stochastic simulations for extensions of our model system showing that the stochastic yield catastrophe indeed persists. We start by considering the ring model from the previous section but take polymer-polymer binding into account in addition to growth via monomer attachment (*Figure 6*). In detail, we assume that two structures of arbitrary size (and with combined length $\leq L$) bind at rate $\nu$ if they fit together, that is if the left (right) end of the first structure is periodically continued by the right (left) end of the second one. Realistically, the rate of binding between two structures is expected to decrease with the motility and thus the sizes of the structures. In order to assess the effect of polymer-polymer binding, we focus on the worst case where the rate for binding is independent of the size of both structures. If a stochastic yield catastrophe occurs for this choice of parameters, we expect it to be even more pronounced in all the 'intermediate cases'. *Figure 6* shows the dependence of the yield on the activation rate in the polymer-polymer model. As before, yield increases below a critical activation rate and then saturates at an imperfect value for small activation rates. Decreasing the number of particles per species, decreases this saturation value. Compared to the original model, the stochastic yield catastrophe is mitigated but still significant: For structures of size $S = L = 100$, yield saturates at around 0.87 for $N = 100$ particles per species and at around 0.33 for $N = 10$ particles per species. We thus conclude that polymer-polymer binding indeed alleviates the stochastic yield catastrophe but does not resolve it. Since binding only happens between consecutive species, structures with overlapping parts intrinsically can not bind together and depletion traps continue to occur. Taken together, also in the extended model, fluctuations in the availability of the different species lead to an excess of intermediate-sized structures that get kinetically trapped due to structural mismatches. Note that in the extreme case of $N = 1$, incomplete polymers can always combine into one final ring structure so that in this case the yield is always 1. Analogously, for high activation rates yield is improved for $N = 10$ compared to $N \geq 50$ (*Figure 6b*).

Kinetic trapping due to structural mismatches can occur in every (partially) irreversible heterogeneous assembly process with finite-sized target structure and limited resources. From our results, we thus expect a stochastic yield catastrophe to be common to such systems. In order to further test this hypothesis, we simulated another variant of our model where finite sized squares assemble via monomer attachment from a pool of initially inactive particles, see *Figure 7*. In contrast to the

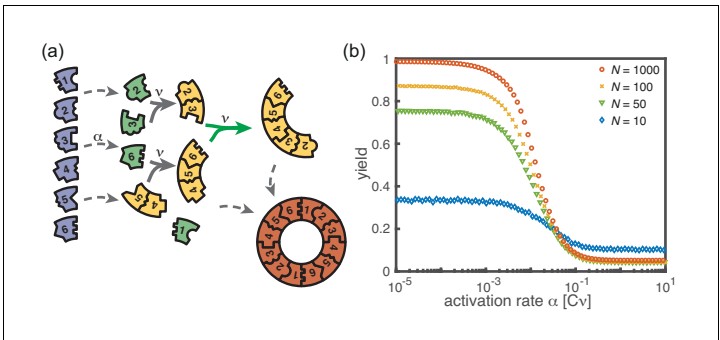

**Figure 6.** Extended model including polymer-polymer binding. (a) In the extended model, structures not only grow by monomer attachment but also by binding with another polymer (colored arrow). As before, binding only happens between periodically consecutive species with rate $\nu$ per binding site. So, the reaction rate for two polymers is identical to the one for monomer-polymer binding, $\nu$. Furthermore, only polymers with combined length $\leq L$ can bind. All other processes and rules are the same as in the original model described in *Figure 1*. (b) The yield of the extended model as obtained from stochastic simulations is shown in dependence of the activation rate $\alpha$ for $S = L = 100$, $\mu = \nu = 1$, $L_{\mathrm{nuc}} = 2$ and different values of the number of particles per species, $N$ (averaged over 1024 ensembles). The qualitative behavior is the same as for the original model. In particular, yield saturates (in the stochastic limit) at an imperfect value for slow activation rates. Note that for small particle numbers polymer-polymer binding results in an increase of the minimal yield (here for large activation rates). This is due to the fact that even in the case where a priori too many nucleation events happen, polymers can combine into final structures.

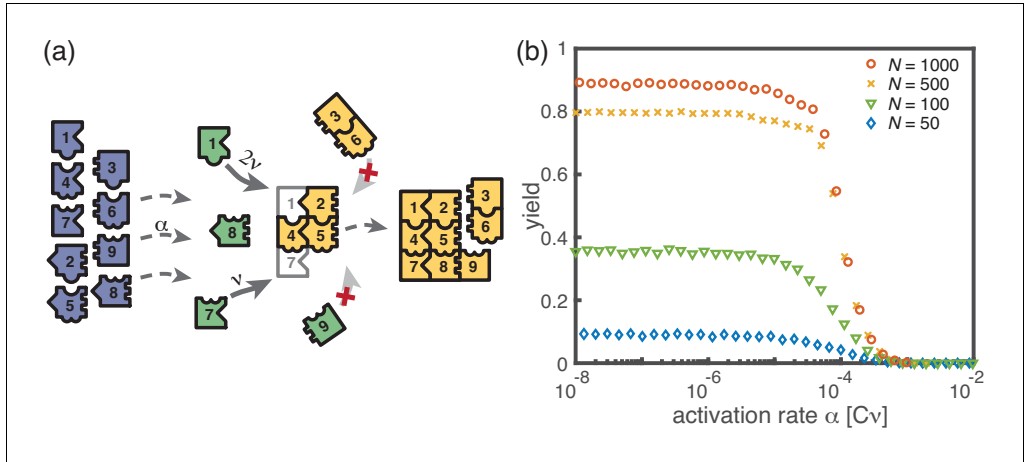

**Figure 7.** Assembly of squares of size $\sqrt{L} \times \sqrt{L}$ from $L$ different particle species. (a) As in the ring models, there are $N$ monomers of each species in the system. All particles are initially in an inactive state (blue) and are activated at the same per-capita rate $\alpha$. Once active (green), species with neighboring position within the square (left/right, up/down) can bind to each other. Structures grow by attachment of single monomers until the square is complete (absorbing state). Depending on the number $b$ of contacts between the monomer and the structure, the corresponding rate is $b\nu$. For simplicity, all polymers (yellow) are stable ($L_{\mathrm{nuc}} = 2$) and we do not consider polymer-polymer binding. (b) The yield of the square model as obtained from stochastic simulations is shown in dependence of the activation rate $\alpha$ for $S = L = 100$, $\mu = \nu = 1$ and different values of the number of particles per species, $N$ (averaged over 256 ensembles). The qualitative behavior is the same as for the previous models: Whereas the yield is poor for large activation rates, it strongly increases below a threshold value and saturates (in the stochastic limit) at an imperfect value < 1 for small activation rates. The saturation value decreases with decreasing number of particles in the system.

original model, the assembled structures are non-periodic and exhibit a non-linear assembly path where structures can grow independently in two dimensions. While the ring model assumes a sequential order of binding of the monomers, the square allows for a variety of distinct assembly paths that all lead to the same final structure. Note that, because of the absence of periodicity, the square model is only well defined for the completely heterogeneous case. *Figure 7* depicts the dependence of the yield on the activation rate for a square of size $S = 100$. Also in this case, we find that the yield saturates at an imperfect value for small activation rates. Hence, we showed that the stochastic yield catastrophe is not resolved neither by accounting for polymer-polymer combination nor by considering more general assembly processes with multiple parallel assembly paths. This observation supports the general validity of our findings and indicates that stochastic yield catastrophes are a general phenomenon of (partially) irreversible and heterogeneous self-assembling systems that occur if particle number fluctuations are non-negligible.

## Discussion

Our results show that different ways to slow down nucleation are indeed not equivalent, and that the explicit implementation is crucial for assembly efficiency. Susceptibility to stochastic effects is highly dependent on the specific scenario. Whereas systems for which dimerization limits nucleation are robust against stochastic effects, stochastic yield catastrophes can occur in heterogeneous systems when resource supply limits nucleation. The occurrence of stochastic yield catastrophes is not captured by the deterministic rate equations, for which the qualitative behavior of both scenarios is the same. Therefore, a stochastic description of the self-assembly process, which includes fluctuations in the availability of the different species, is required. The interplay between stochastic and deterministic dynamics can lead to a plethora of interesting behaviors. For example, the combination of slow activation and slow nucleation may result in a non-monotonic dependence of the yield on the activation rate. While deterministically, yield is always improved by decreasing the activation rate, stochastic fluctuations between species strongly suppress the yield for small activation rate by

effectively enhancing the nucleation speed. This observation clearly demonstrates that a *deterministically* slow nucleation speed is not sufficient in order to obtain good yield in heterogeneous self-assembly. For example, a slow activation step does not necessarily result in few nucleation events although deterministically this behavior is expected. Thus, our results indicate that the slow nucleation principle has to be interpreted in terms of the stochastic framework and have important implications for yield optimization.

We showed that demographic noise can cause stochastic yield catastrophes in heterogeneous self-assembly. However, other types of noise, such as spatiotemporal fluctuations induced by diffusion, are also expected to trigger stochastic yield catastrophes. Hence, our results have broad implications for complex biological and artificial systems, which typically exhibit various sources of noise. We characterize conditions under which stochastic yield catastrophes occur, and demonstrate how they can be mitigated. These insights could usefully inform the design of experiments to circumvent yield catastrophes: In particular, while slow provision of constituents is a feasible strategy for experiments, it is highly susceptible to stochastic effects. On the other hand, irrespective of its robustness to stochastic effects, the experimental realization of the dimerization scenario relies on cooperative or allosteric effects in binding, and may therefore require more sophisticated design of the constituents (*Sacanna et al., 2010*; *Zeravcic et al., 2017*). Our theoretical analysis shows that stochasticity can be alleviated either by decreasing heterogeneity (presumably lowering realizable complexity) or by increasing reversibility (potentially requiring fine-tuning of bond strengths and reducing the stability of the assembly product). Alternative approaches to control stochasticity include the promotion of specific assembly paths (*Murugan et al., 2015*; Gartner, Graf and Frey, in preparation) and the control of fluctuations (Graf, Gartner and Frey, in preparation). One possibility to test these ideas and the ensuing control strategies could be via experiments based on DNA origami. Instead of building homogeneous ring structures as in *Wagenbauer et al. (2017)*, one would have to design heterogeneous ring structures made from several different types of constituents with specified binding properties. By varying the opening angle of the 'wedges' (and thus the preferred number of building blocks in the ring) and/or the number of constituents, both the target structure size $L$ as well as the heterogeneity of the target structure $S$ could be controlled.

Moreover, the ideas presented in this manuscript are relevant for the understanding of intracellular self-assembly. In cells, provision of building blocks is typically a gradual process, as synthesis is either inherently slow or an explicit activation step, such as phosphorylation, is required. In addition, the constituents of the complex structures assembled in cells are usually present in small numbers and subject to diffusion. Hence, stochastic yield catastrophes would be expected to have devastating consequences for self-assembly, unless the relevant cellular processes use elaborate control mechanisms to circumvent stochastic effects. Further exploration of these control mechanisms should enhance the understanding of self-assembly processes in cells and help improve synthesis of complex nanostructures.

## Materials and methods

All our simulation data was generated with either C++ or MATLAB. The source code is available at the eLife website.

Here we show the derivation of *Equation 1* in the main text, giving the threshold values for the rate constants below which finite yield is obtained. The details can be found in Appendices 1–3.

### Master equation and chemical rate equations

We start with the general Master equation and derive the chemical rate equations (deterministic/mean-field equations) for the heterogeneous self-assembly process. We renounce to show the full Master equation here but instead state the system that describes the evolution of the first moments. To this end, we denote the random variable that describes the number of polymers of size $\ell$ and species $s$ in the system at time $t$ by $n_\ell^s(t)$ with $2 \leq \ell < L$ and $1 \leq s \leq S$. The species of a polymer is defined by the species of the respective monomer at its left end. Furthermore, $n_0^s$ and $n_1^s$ denote the number of inactive and active monomers of species $s$, respectively, and $n_L$ the number of complete rings. We signify the reaction rate for binding of a monomer to a polymer of size $\ell$ by $\nu_\ell$. $\alpha$ denotes the activation rate and $\delta_\ell$ the decay rate of a polymer of size $\ell$. By $\langle ... \rangle$ we indicate (ensemble)

averages. The system governing the evolution of the first moments (the averages) of the $\{n_\ell^s\}$ is then given by:

$$\frac{d}{dt}\langle n_0^s \rangle = -\alpha \langle n_0^s \rangle, \tag{2a}$$

$$\frac{d}{dt}\langle n_1^s \rangle = \alpha \langle n_0^s \rangle - \sum_{\ell=1}^{L-1} \nu_\ell \left( \langle n_1^s n_\ell^{s+1} \rangle + \langle n_1^s n_\ell^{s-\ell} \rangle \right) + \sum_{\ell=2}^{L_{\mathrm{nuc}}-1} \sum_{k=s+1-\ell}^{k=s} \delta_\ell \langle n_\ell^k \rangle, \tag{2b}$$

$$\frac{d}{dt}\langle n_2^s \rangle = \nu_1 \langle n_1^s n_1^{s+1} \rangle - \nu_2 \langle n_2^s n_1^{s+2} \rangle - \nu_2 \langle n_2^s n_1^{s-1} \rangle - \delta_2 \langle n_2^s \rangle \mathbf{1}_{\{2<L_{\mathrm{nuc}}\}}, \tag{2c}$$

$$\frac{d}{dt}\langle n_\ell^s \rangle = \nu_{\ell-1} \langle n_{\ell-1}^s n_1^{\ell+s-1} \rangle + \nu_{\ell-1} \langle n_{\ell-1}^{s+1} n_1^s \rangle - \nu_\ell \langle n_\ell^s n_1^{s+\ell} \rangle - \nu_\ell \langle n_\ell^s n_1^{s-1} \rangle - \delta \langle n_\ell^s \rangle \mathbf{1}_{\{\ell<L_{\mathrm{nuc}}\}}, \tag{2d}$$

$$\frac{d}{dt}\langle n_L^s \rangle = \nu_{L-1} \langle n_{L-1}^s n_1^{L+s-1} \rangle + \nu_{L-1} \langle n_{L-1}^{s+1} n_1^s \rangle. \tag{2e}$$

The different terms of this equation are illustrated graphically in *Figure 8*. The first equation describes loss of inactive particles due to activation at rate $\alpha$. *Equation 2b* gives the temporal change of the number of active monomers that is governed by the following processes: activation of inactive monomers at rate $\alpha$, binding of active monomers to the left or to the right end of an existing structure of size $\ell$ at rate $\nu_\ell$, and decay of below-critical polymers of size $\ell$ into monomers at rate $\delta_\ell$ (disassembly). *Equations 2c and 2d* describe the dynamics of dimers and larger polymers of size $3 \leq \ell < L$, respectively. The terms account for reactions of polymers with active monomers (polymerization) as well as decay in the case of below-critical polymers (disassembly). The indicator function $\mathbf{1}_{\{x<L_{\mathrm{nuc}}\}}$ equals 1 if the condition $x<L_{\mathrm{nuc}}$ is satisfied and 0 otherwise. Note that a polymer of size $\ell \geq 3$ can grow by attaching a monomer to its left or to its right end whereas the formation of a dimer of a specific species is only possible via one reaction pathway (dimerization reaction). Finally, polymers of length $L$ – the complete ring structures – form an absorbing state and, therefore, include only the respective gain terms (cf *Equation 2e*).

We simulated the Master equation underlying *Equation 2* stochastically using Gillespie's algorithm. For the following deterministic analysis, we neglect correlations between particle numbers $\{n_\ell^s\}$, which is valid assumption for large particle numbers. Then the two-point correlator can be approximated as the product of the corresponding mean values (mean-field approximation)

$$\langle n_i^s n_j^k \rangle = \langle n_i^s \rangle \langle n_j^k \rangle \ \forall s,k \tag{3}$$

Furthermore, for the expectation values it must hold

$$\langle n_\ell^s \rangle = \langle n_\ell^1 \rangle \ \forall s \tag{4}$$

because all species have equivalent properties (there is no distinct species) and hence the system is invariant under relabelling of the upper index. By

$$c_\ell := \frac{\langle n_\ell^s \rangle}{V}, \tag{5}$$

we denote the concentration of any monomer or polymer species of size $\ell$, where $V$ is the reaction volume. Due to the symmetry formulated in *Equation 4*, the heterogeneous assembly process decouples into a set of $S$ identical and independent homogeneous assembly processes in the deterministic limit. The corresponding homogeneous system then is described by the following set of equations that is obtained by applying (*Equation 3*, *Equation 4*) and (*Equation 5*) to (*Equation 2*)

$$\frac{d}{dt}c_0 = -\alpha c_0, \tag{6a}$$

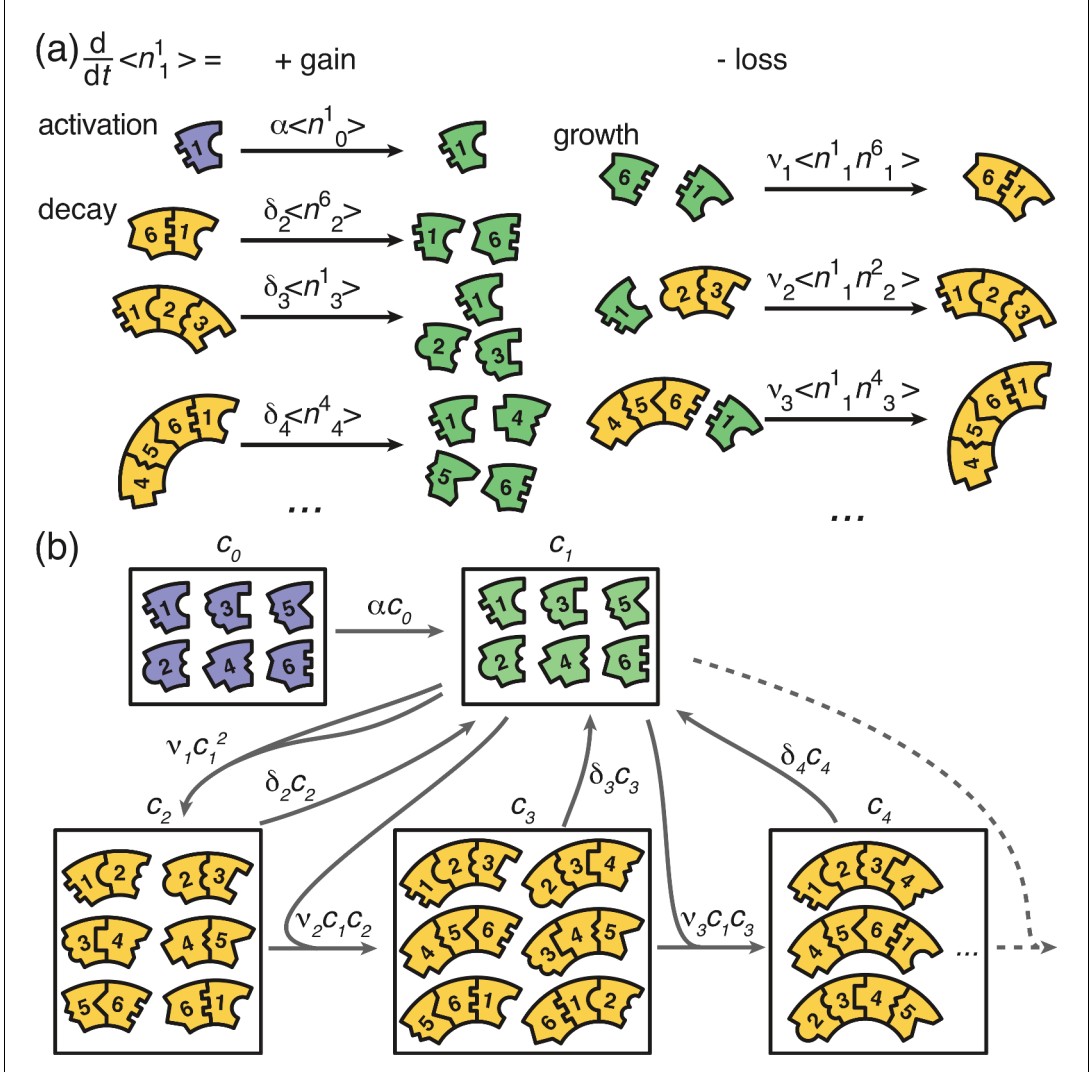

**Figure 8.** Graphical illustration of *Equations 2 and 6*. (a) Visualization of the gain and loss terms in the dynamics of the active monomers in *Equation 2b*. Gain of active monomers is due to activation of inactive monomers as well as decay of unstable polymers. Loss of active monomers is due to dimerization and attachment of monomers to larger polymers. (b) Visualization of the transitions between clusters of different sizes (without distinction of species). The first and second box represent the inactive and active monomers in the system, the subsequent boxes each represent the ensemble of polymers of a certain size. The arrows between the boxes show possible reactions and transitions with the reaction rates indicated accordingly. Each arrow starting from or leading to a box is associated with a corresponding loss or gain term on the right hand side of *Equation 2* and *Equation 6*.

$$\frac{d}{dt}c_1 = \alpha c_0 - 2c_1 \sum_{\ell=1}^{L-1} \nu_\ell c_\ell + \sum_{\ell=2}^{L_{\mathrm{nuc}}-1} l \delta_\ell c_\ell, \tag{6b}$$

$$\frac{d}{dt}c_2 = \nu_1 c_1^2 - 2\nu_2 c_1 c_2 - \delta_2 c_2 \mathbf{1}_{\{2<L_{\mathrm{nuc}}\}}, \tag{6c}$$

$$\frac{d}{dt}c_\ell = 2\nu_{\ell-1} c_1 c_{\ell-1} - 2\nu_\ell c_1 c_\ell - \delta_\ell c_\ell \mathbf{1}_{\{\ell<L_{\mathrm{nuc}}\}}, \qquad \text{for } 3 \leq \ell < L, \tag{6d}$$

$$\frac{d}{dt}c_L = 2\nu_{L-1} c_1 c_{L-1}. \tag{6e}$$

The rate constants $\nu_\ell$ in *Equations 6 and 2* differ by a factor of $V$. For convenience, we use however the same symbol in both cases. The rate constants $\nu_\ell$ in *Equation 6* can be interpreted in the usual units $\left[\frac{\text{liter}}{\text{molsec}}\right]$. Due to the symmetry, the yield, which is given by the quotient of the number of completely assembled rings and the maximum number of complete rings, becomes independent of the number of species $S$

$$\text{yield(t)} = \frac{S c_L(t) V}{S N L^{-1}} = \frac{c_L(t) V L}{N}. \tag{7}$$

Hence, it is enough to study the dynamics of the homogeneous system, *Equation 6*, to identify the condition under which non zero yield is obtained.

## Effective description by an advection-diffusion equation

The dynamical properties of the evolution of the polymer-size distribution become evident if the set of ODEs, *Equation 6*, is rewritten as a partial differential equation. This approach was previously described in the context of virus capsid assembly (*Zlotnick et al., 1999*; *Morozov et al., 2009*). For simplicity, we restrict ourselves to the case $L_{\text{nuc}} = 2$ and let $\nu_1 = \mu$ and $\nu_{\ell \geq 2} = \nu$. Then, for the polymers with $\ell > 2$ we have

$$\partial_t c_\ell = 2\nu c_1 \left[ c_{\ell-1} - c_\ell \right]. \tag{8}$$

As a next step, we approximate the index $\ell \in \{2, 3, \ldots, L\}$ indicating the length of the polymer as a continuous variable $x \in [2, L]$ and define $c(x = \ell) := c_\ell$. By $A := c_1$ we denote the concentration of active monomers in the following to emphasize their special role. Formally expanding the right-hand side of *Equation 8* in a Taylor series up to second order

$$c(\ell - 1) = c(\ell) - \partial_x c(\ell) + \frac{1}{2}\partial_x^2 c(\ell), \tag{9}$$

one arrives at the advection-diffusion equation with both advection and diffusion coefficients depending on the concentration of active monomers $A(t)$

$$\partial_t c(x) = -2\nu A \, \partial_x c(x) + \nu A \, \partial_x^2 c(x). \tag{10}$$

*Equation 10* can be written in the form of a continuity equation $\partial_t c(x) = -\partial_x J(x)$ with flux $J = 2\nu A \, c - \nu A \, \partial_x c$. The flux at the left boundary $x = 2$ equals the influx of polymers due to dimerization of free monomers $J(2, t) = \mu A^2$. This enforces a Robin boundary condition at $x = 2$

$$2\nu A \, c(2, t) - \nu A \, \partial_x c(2, t) = \mu A^2. \tag{11}$$

At $x = L$ we set an absorbing boundary $c(L, t) = 0$ so that completed structures are removed from the system. The time evolution of the concentration of active monomers is given by

$$\partial_t A = \alpha C e^{-\alpha t} - 2\mu A^2 - 2\nu A \int_2^L c(x, t) \, dx. \tag{12}$$

The terms on the right-hand side account for activation of inactive particles, dimerization, and binding of active particles to polymers (polymerization).

Qualitatively, *Equation 10* describes a profile that emerges at $x = 2$ from the boundary condition *Equation 11,* moves to the right with time-dependent velocity $2\nu A(t)$ due to the advection term, and broadens with a time-dependent diffusion coefficient $\nu A(t)$. In Appendices 2–3 we show how the full solution of *Equations 10 and 11* can be found assuming knowledge of $A(t)$. Here, we focus only on the derivation of the threshold activation rate and threshold dimerization rate that mark the onset of non-zero yield. Yield production starts as soon as the density wave reaches the absorbing boundary at $x = L$. Therefore, finite yield is obtained if the sum of the advectively travelled distance $d_{\text{adv}}$ and the diffusively travelled distance $d_{\text{diff}}$ exceeds the system size $L - 2$

$$d_{\text{adv}} + d_{\text{diff}} \geq L - 2. \tag{13}$$

According to *Equation 10*, $d_{\mathrm{adv}} = 2\nu \int_0^\infty A(t)dt$ and $d_{\mathrm{diff}} = \sqrt{2\nu \int_0^\infty A(t)dt}$, giving as condition for the onset of finite yield

$$2\nu \int_0^\infty A(t)dt \stackrel{!}{=} \frac{1}{4}\left(\sqrt{1+4(L-2)}-1\right)^2 \approx L - \sqrt{L}, \tag{14}$$

where the last approximation is valid for large $L$.

In order to obtain $\int_0^\infty A(t)dt$ we derive an effective two-component system that governs the evolution of $A(t)$. To this end, we denote the total number of polymers in *Equation 12* by $B(t) := \int_2^\infty c(x,t)\,dx$ (as long as yield is zero the upper boundary is irrelevant and we can consider $L = \infty$). *Equation 12* then reads

$$\frac{d}{dt}A = \alpha C e^{-\alpha t} - 2\mu A^2 - 2\nu A\,B\,, \tag{15}$$

and the dynamics of $B$ is determined from the boundary condition, *Equation 11*

$$\frac{d}{dt}B = \int_2^\infty \partial_t c(x,t)\,dx = \int_2^\infty -\partial_x J(x,t)\,dx = -\underbrace{J(\infty,t)}_{=0} + J(2,t) = \mu A(t)^2. \tag{16}$$

Measuring $A$ and $B$ in units of the initial monomer concentration $C$ and time in units of $(\nu C)^{-1}$ the equations are rewritten in dimensionless units as

$$\frac{d}{dt}A = \omega e^{-\omega t} - 2\eta A^2 - 2AB, \tag{17a}$$

$$\frac{d}{dt}B = \eta A^2, \tag{17b}$$

where $\omega = \frac{\alpha}{\nu C}$ and $\eta = \frac{\mu}{\nu}$. *Equation 17* describes a closed two-component system for the concentration of active monomers $A$ and the total concentration of polymers $B$. It describes the dynamics exactly as long as yield is zero. In order to evaluate the condition (14) we need to determine the integral over $A(t)$ as a function of $\omega$ and $\eta$

$$\int_0^\infty A_{\omega,\eta}(t)dt := g(\omega,\eta)\,. \tag{18}$$

To that end, we proceed by looking at both scenarios separately. The numerical analysis, confirming our analytic results, is given in Appendix 3.

## Dimerization scenario

The activation rate in the dimerization scenario is $\alpha \to \infty$, and instead of the term $\omega e^{-\omega t}$ in $\mathrm{d}A/\mathrm{d}t$, we set the initial condition $A(0) = 1$ (and $B(0) = 0$). Furthermore, $\eta = \mu/\nu \ll 1$ and we can neglect the term proportional to $\eta$ in $\mathrm{d}A/\mathrm{d}t$. As a result,

$$\frac{\mathrm{d}A}{\mathrm{d}B} = -\frac{2B}{\eta A}.$$

Solving this equation for $A$ as a function of $B$ using the initial condition $A(B=0)=1$, the totally travelled distance of the wave is determined to be

$$2g(\omega,\eta) = 2\frac{\pi}{2\sqrt{2}}\frac{1}{\sqrt{\eta}}, \tag{19}$$

where for the evaluation of the integral we used the substitution $\eta A^2 \mathrm{d}t = \mathrm{d}B$.

## Activation scenario

In the activation scenario, yield sets in only if the activation rate and thus the effective nucleation rate is slow. As a result, in addition to $\omega \ll 1$, we can again neglect the term proportional to $\eta$ in $\mathrm{d}A/\mathrm{d}t$. This time, however, we have to keep the term $\omega e^{-\omega t}$. As a next step, we assume that $\mathrm{d}A/\mathrm{d}t$ is much smaller than the remaining terms on the right-hand side, $\omega e^{-\omega t}$ and $-2AB$. This assumption might seem crude at first sight but is justified a posteriori by the solution of the equation (see Appendix 3). Hence, we get the algebraic equation $A(t) = \omega e^{-\omega t}/(2B(t))$. Using it to solve $\mathrm{d}B/\mathrm{d}t = \eta A^2$ for $B$, and then to determine $A$, the totally travelled distance of the wave is deduced as

$$2g(\omega, \eta) = 2\frac{3^{2/3}\sqrt{\pi}\Gamma(2/3)}{6\Gamma(7/6)}(\omega\eta)^{-1/3}. \tag{20}$$

Taken together, we therefore obtain two conditions out of which one must be fulfilled in order to obtain finite yield

$$2a(\eta\omega)^{-\frac{1}{3}} \geq L - \sqrt{L} \Rightarrow \quad \alpha < \alpha_{\mathrm{th}} := P_\alpha \frac{\nu}{\mu} \frac{\nu C}{(L - \sqrt{L})^3} \tag{21}$$

$$\text{or} \quad 2b\eta^{-\frac{1}{2}} \geq L - \sqrt{L} \Rightarrow \quad \mu < \mu_{\mathrm{th}} := P_\mu \frac{\nu}{(L - \sqrt{L})^2}, \tag{22}$$

where $a$ and $b$ are numerical factors, and $P_\alpha = 8a^3 \approx 5.77$ and $P_\mu = 4b^2 \approx 4.93$. This verifies *Equation 1* in the main text.

## Acknowledgements

We thank Nigel Goldenfeld for a stimulating discussion, and Raphaela Geßele and Laeschkir Hassan for helpful feedback on the manuscript. This research was supported by the German Excellence Initiative via the program 'NanoSystems Initiative Munich'(NIM) and was funded by the Deutsche Forschungsgemeinschaft (DFG, German Research Foundation) under Germany's Excellence Strategy – EXC-2094–390783311. FMG and IRG are supported by a DFG fellowship through the Graduate School of Quantitative Biosciences Munich (QBM). We also gratefully acknowledge financial support by the DFG Research Training Group GRK2062 (Molecular Principles of Synthetic Biology). Finally, EF thanks the Aspen Center for Physics, which is supported by National Science Foundation grant PHY-1607611, for their hospitality and inspiring discussions with colleagues.

## Additional information

### Funding

| Funder | Grant reference number | Author |
| --- | --- | --- |
| Deutsche Forschungsgemeinschaft | GRK2062 | Patrick Wilke |
| Deutsche Forschungsgemeinschaft | QBM | Florian M Gartner<br>Isabella R Graf |
| Aspen Center for Physics | PHY-160761 | Erwin Frey |
| Deutsche Forschungsgemeinschaft | EXC-2094 - 390783311 | Erwin Frey |

The funders had no role in study design, data collection and interpretation, or the decision to submit the work for publication.

### Author contributions

Florian M Gartner, Isabella R Graf, Patrick Wilke, Conceptualization, Data curation, Software, Formal analysis, Validation, Investigation, Visualization, Methodology, Project administration; Philipp M Geiger, Conceptualization, Validation, Investigation, Visualization, Project administration; Erwin Frey,

Conceptualization, Resources, Supervision, Funding acquisition, Validation, Methodology, Project administration

### Author ORCIDs
Florian M Gartner ⬚ https://orcid.org/0000-0002-9801-4288
Isabella R Graf ⬚ https://orcid.org/0000-0001-9169-9109
Erwin Frey ⬚ https://orcid.org/0000-0001-8792-3358

### Decision letter and Author response
Decision letter https://doi.org/10.7554/eLife.51020.sa1
Author response https://doi.org/10.7554/eLife.51020.sa2

## Additional files

### Supplementary files
• Source code 1. C++ code for original ring model without polymer-polymer binding.
• Source code 2. C++ code for extended ring model with polymer-polymer binding.
• Source code 3. C++ code for square model.
• Source code 4. MATLAB code for original ring model.
• Transparent reporting form

### Data availability
All data was generated from stochastic simulations in C++ and deterministic simulations in Matlab. The source code files are included with the article.

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

## Appendix 1

# Chemical reaction equations and the equivalence of models with different numbers of species

In this section we derive the chemical rate equations (deterministic equations) for the self-assembly process as described in the main text. Furthermore, we show that for general $S$ in the deterministic limit the model is equivalent to a set of $S$ independent assembly processes with only one species.

## Homogeneous structures

First, we consider the homogeneous model ($S = 1$). By $c_\ell(t)$ we denote the concentration of complexes of length $\ell$ ($\ell \geq 2$) at time $t$, $c_1(t)$ is the concentration of active monomers and $c_0(t)$ the concentration of inactive monomers at time $t$. In the following we will usually skip the time argument for better readability. We denote the reaction rate for binding of a monomer to a polymer of size $\ell$ by $\nu_\ell$. The model from the main text is recovered by setting $\nu_\ell := \mu_\ell$ if $\ell < L_{\mathrm{nuc}}$, and $\nu_\ell := \nu$ otherwise. The ensuing set of ordinary differential equations then reads:

$$\frac{d}{dt}c_0 = -\alpha\,c_0\,, \tag{A1a}$$

$$\frac{d}{dt}c_1 = \alpha\,c_0 - 2c_1\sum_{\ell=1}^{L-1}\nu_\ell\,c_\ell + \sum_{\ell=2}^{L_{\mathrm{nuc}}-1} l\,\delta_\ell\,c_\ell\,, \tag{A1b}$$

$$\frac{d}{dt}c_2 = \nu_1\,c_1^2 - 2\nu_2\,c_1\,c_2 - \delta_2\,c_2\,\mathbf{1}_{\{2<L_{\mathrm{nuc}}\}}\,, \tag{A1c}$$

$$\frac{d}{dt}c_\ell = 2\nu_{\ell-1}\,c_1\,c_{\ell-1} - 2\nu_\ell\,c_1\,c_\ell - \delta_\ell\,c_\ell\,\mathbf{1}_{\{\ell<L_{\mathrm{nuc}}\}}\,, \qquad \text{for}\,3 \leq \ell < L\,, \tag{A1d}$$

$$\frac{d}{dt}c_L = 2\nu_{L-1}\,c_1\,c_{L-1}\,. \tag{A1e}$$

The indicator function $\mathbf{1}_{\{x<L_{\mathrm{nuc}}\}}$ equals 1 if the condition $x<L_{\mathrm{nuc}}$ is satisfied and 0 otherwise. The first equation describes loss of inactive particles due to activation at rate $\alpha$. It is uncoupled from the remainder of the equations and is solved by $c_0(t) = Ce^{-\alpha t}$, with $C$ denoting the initial concentration of inactive monomers. The temporal change of the active monomers is governed by the following processes (**Equation A1b**): activation of inactive monomers at rate $\alpha$, binding of active monomers to existing structures at rate $\nu_\ell$ (polymerization), and decay of below-critical polymers into monomers at rate $\delta_\ell$ (disassembly). All binding rates appear with a factor of 2 because a monomer can attach to a polymer on its left or on its right end.

Note that there is a subtlety with the dimerization term $2\nu_1 c_1^2$ in **Equation A1b**: the dimerization term as well bears a factor of 2 because two identical monomers $A$ and $B$ can form a dimer in two possible ways, either as $AB$ or $BA$. Additionally, there is a stoichiometric factor of 2 for the monomers in this reaction. However, one factor of 2 is cancelled again because, assuming there are $n$ monomers, the number of ordered pairs of monomers that describe possible reaction partners is $\frac{1}{2}n(n-1) \approx n^2/2$ (if $n$ is large) rather than $n^2$ (the number of reaction partners when two different species react). This leaves us with a single factor of 2 like for all the other binding reactions.

**Equations A1c and A1d** describe the dynamics of dimers and larger polymers of size $3 \leq \ell < L$, respectively. The terms account for reactions of polymers with active monomers (polymerization) as well as decay in the case of below-critical polymers (disassembly). The dimerization term in the equation for $\partial_t c_2$ lacks the factor of 2 because the stoichiometric

factor is missing for the dimers as compared with the dimerization term for the monomers in the line above. Finally, polymers of length $L$ – the complete ring structures – form an absorbing state and therefore only include a reactive gain term (**Equation A1e**).

## Heterogeneous structures

Next we consider systems with more than one particle species ($S>1$). The heterogeneous system can be described by dynamical equations equivalent to the homogeneous system. We show this starting from a full description that distinguishes both monomers and polymers into a set of different species $1, \ldots, S$. The species of a polymer is defined by the species of the respective monomer at its left end. As polymers assemble in consecutive order of species, a polymer is uniquely determined by its length and species (i.e. species of leftmost monomer). In that sense, $c_\ell^s$ with $0 \leq \ell < L$ and $1 \leq s \leq S$ denotes the concentration of a polymer of length $\ell$ and species $s$ ($c_0^s$ and $c_1^s$ again denote inactive and active monomers of species $s$, respectively). For example, $c_4^5$ denotes the concentration of polymers [5678] if $S \geq 8$, or of polymers [5612] if $S = 6$. Upper indices are always assumed to be taken modulo $S$ whenever they lie outside the range $[1, S]$. Therefore, the dynamics of the concentrations $c_\ell^s$ with $3 \leq \ell < L$ is given by

$$\frac{d}{dt}c_\ell^s = \nu_{\ell-1}\, c_{\ell-1}^s\, c_1^{\ell+s-1} + \nu_{\ell-1}\, c_{\ell-1}^{s+1}\, c_1^s - \nu_\ell\, c_\ell^s\, c_1^{s+\ell} - \nu_\ell\, c_\ell^s\, c_1^{s-1} - \delta\, c_\ell^s\, \mathbf{1}_{\{\ell < L_{\mathrm{nuc}}\}}. \tag{A2}$$

The terms on the right-hand side account for the influx due to binding of the respective polymers of length $\ell - 1$ with a monomer either on the right or on the left (first and second term), and for the outflux due to reactions of a polymer of length $\ell$ and species $s$ with a monomer on the right or on the left (third and fourth term), as well as for decay into monomers for $\ell < L_{\mathrm{nuc}}$ (last term). For the dynamics of the dimers, however, there is only one gain term arising from dimerization:

$$\frac{d}{dt}c_2^s = \nu_1\, c_1^s\, c_1^{s+1} - \nu_2\, c_2^s\, c_1^{s+2} - \nu_2\, c_2^s\, c_1^{s-1} - \delta_2\, c_2^s\, \mathbf{1}_{\{2 < L_{\mathrm{nuc}}\}}. \tag{A3}$$

Equivalently, for the active monomers we find:

$$\frac{d}{dt}c_1^s = \alpha\, C e^{-\alpha t} - c_1^s \sum_{\ell=1}^{L-1} \nu_\ell \left( c_\ell^{s+1} + c_\ell^{s-\ell} \right) + \sum_{\ell=2}^{L_{\mathrm{nuc}}-1} \sum_{k=s+1-\ell}^{k=s} \delta_\ell c_\ell^k.$$

Now we exploit the symmetry of the system with respect to the species index, that is, the upper index in $\{c_\ell^s\}$: Since all species in the system are equivalent, the dynamic equations are invariant under relabelling of the upper indices. Consequently, it must hold that:

$$c_\ell^s(t) = c_\ell^k(t), \quad \text{for any } s, k \leq S \text{ at any time } t. \tag{A5}$$

In other words, the upper index is irrelevant and can also be discarded. The variable $c_\ell$ then denotes the concentration of *any* one polymer species of length $\ell$. Taking advantage of this symmetry for the equations of the heterogeneous system, (**Equation A2**, **Equation A3** and **Equation A4**), and collecting equal terms leads to a set of equations fully identical to those for the homogeneous system (**Equation A1**). We show the equivalence to the homogeneous model exemplarily for the dynamics of the polymers with size $\ell \geq 3$ in **Equation A2**. Applying $c_\ell^s(t) = c_\ell(t)$ to **Equation A2** yields for the dynamics of the concentration of an arbitrary polymer species of size $\ell$:

$$\begin{aligned}\frac{d}{dt}c_\ell &= \nu_{\ell-1}\, c_{\ell-1}\, c_1 + \nu_{\ell-1}\, c_{\ell-1}\, c_1 - \nu_\ell\, c_\ell\, c_1 - \nu_\ell\, c_\ell\, c_1 - \delta\, c_\ell\, \mathbf{1}_{\{\ell < L_{\mathrm{nuc}}\}}. \\ &= 2\nu_{\ell-1}\, c_{\ell-1}\, c_1 - 2\nu_\ell\, c_\ell\, c_1 - \delta\, c_\ell\, \mathbf{1}_{\{\ell < L_{\mathrm{nuc}}\}},\end{aligned}$$

which is identical to the respective dynamic **Equation A1d** for the homogeneous model. The other equations for the heterogeneous system reduce to those for the homogeneous system in an analogous manner.

Summarizing, we have shown that the (deterministic) heterogeneous assembly process decouples into a set of $S$ identical and independent homogeneous processes. In particular, yield, which is given by the quotient of the number of completely assembled rings and the maximal possible number of complete rings, becomes independent of $S$:

$$\text{yield}(t) = \frac{S c_L(t)}{S N L^{-1}} = \frac{c_L(t) L}{N} \ . \tag{A6}$$

## Appendix 2

### Effective description of the evolution of the polymer size distribution as an advection-diffusion equation

The dynamical properties of the evolution of the polymer size distribution become evident if the set of ODEs, *Equation 1,* is rewritten as a partial differential equation. This approach was previously described in the context of virus capsid assembly (*Morozov et al., 2009*; *Zlotnick et al., 1999*; *Endres and Zlotnick, 2002*) but we will restate the essential steps here for the convenience of the reader. To this end we interpret the length index of the polymer $\ell \in \{2, 3, \ldots, L\}$ as a continuous variable that we rename $x \in [2, L]$. With such a continuous description in view we write $c(x = \ell) := c_\ell$ to denote the concentration of polymers of size $\ell$.

Since the active monomers play a special role, we denote their concentration in the following by $A$. For simplicity we restrict our discussion to the case $L_{\mathrm{nuc}} = 2$ and let $\nu_1 = \mu$ and $\nu_{\ell \geq 2} = \nu$. Generalizations to $L_{\mathrm{nuc}} > 2$ can be done in a similar way. Then, for the polymers with $\ell \geq 3$ we have:

$$\partial_t c(\ell) = 2\nu A \left[ c(\ell - 1) - c(\ell) \right]. \tag{A7}$$

Formally, expanding the right-hand side in a Taylor series up to second order

$$c(\ell - 1) = c(\ell) - \partial_x c(\ell) + \frac{1}{2} \partial_x^2 c(\ell), \tag{A8}$$

we arrive at an advection-diffusion equation with both advection and diffusion coefficients depending on the concentration of active monomers $A(t)$,

$$\partial_t c(x) = -2\nu A \, \partial_x c(x) + \nu A \, \partial_x^2 c(x). \tag{A9}$$

*Equation A9* can be written in the form of a continuity equation $\partial_t c(x) = -\partial_x J(x)$ with flux $J = 2\nu A \, c - \nu A \, \partial_x c$. The flux at the left boundary, $x = 2$, equals the influx of polymers due to dimerization of free monomers, $J(2, t) = \mu A^2$. This enforces a Robin boundary condition at $x = 2$,

$$2\nu A \, c(2, t) - \nu A \, \partial_x c(2, t) = \mu A^2. \tag{A10}$$

At $x = L$, we have an absorbing boundary $c(L, t) = 0$ so that completed structures are removed from the system. Furthermore, the time evolution of the concentration of active particles is given by

$$\partial_t A = \alpha C e^{-\alpha t} - 2\mu A^2 - 2\nu A \int_2^L c(x, t) \, dx. \tag{A11}$$

The terms on the right-hand side account for activation of inactive particles, dimerization, and binding of active particles to polymers (polymerization).

Qualitatively, *Equation A9* describes a profile that emerges at $x = 2$ from the boundary condition, *Equation A10*, moves to the right with time dependent velocity $2\nu A(t)$ due to the advection term, and broadens with a time-dependent diffusion coefficient $\nu A(t)$. The concentration of active particles $A$ determines both the influx of dimers at $x = 2$, as well as the speed and diffusion of the wave profile.

Next, we derive an expression that solves *Equation A9*, assuming that we know $A(t)$. We start by solving *Equation A9* at the left boundary $c(2, t)$, and then translate the resulting expression to obtain a solution for $c(x, t)$. To obtain $c(2, t)$ in dependence of $a(t)$ we can solve $\frac{d}{dt} c(2, t) = \mu A^2 - 2\nu A c(2, t)$ (see *Equation A1c*) by 'variation of the constants' as

$$c(2,t) = \int_0^t \mu A(\tilde{t})^2 \exp\left[-2\int_{\tilde{t}}^t \nu A(t')dt'\right] d\tilde{t}. \tag{A12}$$

With help of this expression we find $c(x,t)$: Given $c(2,t)$, the advective part of **Equation A9**,

$$\partial_t \tilde{c}(x) = -2\nu A \partial_x \tilde{c}(x). \tag{A13}$$

is solved by

$$c_{\text{advec}}(x,t) = c(2,\tau(x,t)). \tag{A14}$$

Here, $\tau(x,t)$ denotes the time when a particle now at position $x$ and time $t$ was at $x = 2$. In other words, a particle at time $t$ and position $x$ has entered the system at $x = 2$ at time $\tau(x,t)$. This ansatz solves the PDE (**Equation A13**) if and only if $\tau(x,t)$ satisfies

$$\tau(x,t) = \tilde{A}^{-1}\left(\tilde{A}(t) - \frac{x-2}{2\nu}\right) \tag{A15}$$

with $\tilde{A}$ being an arbitrary integral of $A$ such that $\partial_t \tilde{A}(t) = A(t)$ and $\tilde{A}^{-1}$ denoting its inverse. More easily, we find this form of $\tau$ by requiring that the integral over the velocity from time $\tau$ to $t$ equals the travelled distance $x - 2$:

$$\int_\tau^t 2\nu A(t')dt' = x - 2. \tag{A16}$$

To include the diffusive contribution in **Equation A13**, we use the diffusion kernel,

$$k(x,y,t) = \left(4\pi\int_{\tau(y,t)}^t D(t)\right)^{-1/2} \exp\left(\frac{-x^2}{4\int_{\tau(y,t)}^t D(t)}\right), \tag{A17}$$

with the time dependent diffusion constant $D(t) = \nu A(t)$. The kernel $k(x,y,t)$ accounts for the mass that has been diffusively transported from $y$ over a distance of $x$. Because the mass has entered the system at $x = 2$ at time $\tau(y,t)$, it diffused for the time $t - \tau(y,t)$. The complete expression for $c(x,t)$ is then obtained as the convolution of $c_{\text{advec}}(x,t)$ (**Equation A14**), that is obtained from **Equation A12** and **Equation A15**, and the diffusion kernel $k(x,y,t)$ (**Equation A17**):

$$c(x,t) = \int c_{\text{advec}}(s,t)k(x-s,s,t)ds = \int c(2,\tau(s,t))k(x-s,s,t)ds. \tag{A18}$$

Interpreting the terms in the equations and the general form of the solution, we are able to understand the qualitative behavior of the system. If both the activation and the dimerization rate are large, the system produces zero yield: both advection and diffusion are driven by the concentration of active monomers $A$. If activation is fast, the concentration of active monomers $A$ will become large initially since activation is faster than the reaction dynamics. Consequently, provided $\mu \sim \nu$, dimerization dominates over binding because it depends quadratically on $A$, see **Equation A11**. The reservoir of free particles then depletes quickly and cannot sustain the motion of the wave for long enough to reach the absorbing boundary, resulting in a very low yield. Only if either the activation rate is low enough or if $\mu \ll \nu$, the motion of the wave can be sustained until it reaches the absorbing boundary.

## Appendix 3

### Threshold values for the activation and dimerization rate

Based on the analysis from the previous section, we will now determine the threshold activation rate and threshold dimerization rate which mark the onset of non-zero yield. Yield production starts as soon as the density wave reaches the absorbing boundary at $x = L$. Therefore, finite yield is obtained if and only if the sum of the advectively travelled distance $d_{\mathrm{adv}}$ and the diffusively travelled distance $d_{\mathrm{diff}}$ exceeds the system size $L - 2$:

$$d_{\mathrm{adv}} + d_{\mathrm{diff}} \geq L - 2. \tag{A19}$$

The condition for the onset of non-zero yield is obtained by assuming equality in this relation. The advectively travelled distance is obtained from **Equation A16** by setting the borders of the integral over the velocity to $\tau = 0$ and $t = \infty$:

$$d_{\mathrm{adv}} = \int_0^\infty 2\nu A(t')dt'. \tag{A20}$$

The diffusively travelled distance is approximately given by the standard deviation of the Gaussian diffusion kernel, **Equation A17**, again with $\tau = 0$ and $t = \infty$,

$$d_{\mathrm{diff}} = \sqrt{2\nu \int_0^\infty A(t)dt}. \tag{A21}$$

Taken together, we obtain a condition for the onset of finite yield:

$$2\nu \int_0^\infty A(t)dt + \sqrt{2\nu \int_0^\infty A(t)dt} = L - 2. \tag{A22}$$

Substituting $y = \sqrt{2\nu \int A}$ and requiring that $y$ is positive, we solve the quadratic equation and find that **Equation A22** is equivalent to

$$2\nu \int_0^\infty A(t)dt = y^2 = \frac{1}{4}\left(\sqrt{1 + 4(L-2)} - 1\right)^2 \approx L - \sqrt{L}, \tag{A23}$$

where the last approximation is valid for large $L$.

We determine the threshold values for the activation rate $\alpha$ and the dimerization rate $\mu$ by finding solutions of the dynamical equation for the active particles $A(t)$, **Equation A11**, such that the condition, **Equation A23**, is fulfilled. Thus, we start by deriving the dependence of $\int_0^\infty A(t)dt$ on $\alpha$ and $\mu$.

The concentration $c(x, t)$ appears in **Equation A11** only in terms of an integral $\int_2^L c(x, t)\,dx$, counting the total number of polymers in the system. As long as yield is zero there is no outflux of polymers at the absorbing boundary $x = L$ and the total number of polymers in the system only increases due to the influx at the left boundary $x = 2$. As long as yield is zero we can therefore equivalently consider the limit $L \to \infty$. We denote the total number of polymers in **Equation A11** by $B(t) := \int c(x, t)\,dx$ for which the dynamics is determined from the boundary condition, **Equation A10**:

$$\frac{d}{dt}B = \int_2^\infty \partial_t c(x, t)\,dx = \int_2^\infty -\partial_x J(x, t)\,dx = -\underbrace{J(\infty, t)}_{=0} + J(2, t) = \mu A(t)^2. \tag{A24}$$

Hence, as long as yield is zero, the total number of polymers increases with the rate of the

dimerization events. The system then simplifies to a set of two coupled ordinary differential equations for $A$ and $B$:

$$\frac{d}{dt}A = \alpha C e^{-\alpha t} - 2\mu A^2 - 2\nu A B \tag{A25a}$$

$$\frac{d}{dt}B = \mu A^2. \tag{A25b}$$

The dynamics of $A$ and $B$ is equivalent to a two-state activator-inhibitor system, where $A$ dimerizes into $B$ at rate $\mu$, and $B$ degrades (inhibits) $A$ at rate $2\nu$. Note that **Equation A25** describes the exact dynamics of the active monomers $A$ and total number of polymers $B$ in the deterministic system as long as yield is zero. The system has therefore been greatly reduced from originally $SN$ coupled ODEs to now only two coupled ODEs.

For the further analysis it is useful to non-dimensionalize **Equation A25** by measuring $A$ and $B$ in units of the initial concentration of inactive monomers $C$ and time in units of $(\nu C)^{-1}$:

$$\frac{d}{dt}A = \omega e^{-\omega t} - 2\eta A^2 - 2AB, \tag{A26a}$$

$$\frac{d}{dt}B = \eta A^2, \tag{A26b}$$

with the remaining dimensionless parameters $\omega = \frac{\alpha}{\nu C}$ and $\eta = \frac{\mu}{\nu}$. We are interested in the integral over $A(t)$ as a function of $\omega$ and $\eta$,

$$\int_0^\infty A_{\omega,\eta}(t)dt := g(\omega, \eta), \tag{A27}$$

which relates to the totally travelled distance of the wave. Note that, in case of zero yield, $2g(\omega, \eta)$ is the total advectively travelled distance of the wave (cf. **Equation A20**) and the square of the diffusively travelled distance (cf. **Equation A21**).

## Analysis of the dimerization scenario

The dimerization scenario is characterized by fast activation $\alpha \gg C\nu$ and slow dimerization $\mu \ll \nu$. For the dimensionless parameters these assumptions translate to $\eta \ll 1$ and $\eta \ll \omega$. Because for small $\eta \ll 1$ nucleation is much slower than growth we neglect the dimerization term in **Equation A26a** against the growth term. Furthermore, because $\eta \ll \omega$ activation happens on a fast time scale compared with nucleation and we may therefore integrate out the fast time scale assuming that all particles are activated instantaneously at the beginning. The system **Equation A26** then reduces to

$$\frac{d}{dt}A = -2AB, \tag{A28a}$$

$$\frac{d}{dt}B = \eta A^2, \tag{A28b}$$

with the initial condition $A(0) = 1$ and $B(0) = 0$. We divide the first equation by the second one (formally applying the chain rule and the inverse function theorem) to obtain a single equation for the dynamics of $A(B)$:

$$\frac{dA}{dB} = -\frac{2B}{\eta A}, \tag{A29}$$

where $A(B=0) = 1$. This first order ODE can be solved by separation of variables and subsequent integration, yielding

$$A(B) = \sqrt{1 - \frac{2}{\eta}B^2} \,. \tag{A30}$$

Because the number of active monomers $A(t)$ must vanish for $t \to \infty$, the final value of $B$ is

$$B_\infty := B(t{=}\infty) = \sqrt{\frac{\eta}{2}} \,. \tag{A31}$$

Thereby, we calculate the function $g(\eta)$ via variable substitution $dt = \frac{dB}{\eta A^2}$:

$$g(\eta) = \int_0^\infty A(t)dt = \int_0^{B_\infty} A(B)\frac{dB}{\eta A(B)^2} = \frac{1}{\eta}\int_0^{B_\infty} \frac{dB}{\sqrt{1-\frac{2}{\eta}B^2}} = \frac{\pi}{2\sqrt{2}}\,\eta^{-\frac{1}{2}} \,. \tag{A32}$$

So, the dependence of the travelled distance of the wave on $\eta$ obeys a power law with exponent $-\frac{1}{2}$, confirming the previous result (**Morozov et al., 2009**). For the coefficient we find $\frac{\pi}{2\sqrt{2}} \approx 1.1107$.

Additionally, we can determine the time dependent solutions $A(t)$ and $B(t)$. Using the solution for $A(B)$ from **Equation A30** in **Equation A28b** we obtain $B(t)$ as

$$B(t) = \sqrt{\frac{\eta}{2}}\tanh\left(\sqrt{2\eta}t\right) \,. \tag{A33}$$

We use this expression for $B(t)$ in **Equation A28a** to obtain $A(t)$. The resulting ODEs can again be solved by separation of variables as

$$A(t) = \frac{1}{\cosh\left(\sqrt{2\eta}t\right)} \,. \tag{A34}$$

## Analysis of the activation scenario

In the activation scenario, $\alpha \ll C\nu$, such that $\omega \ll 1$ and $\omega \ll \eta$. As we know already that decreasing $\omega$ will slow down nucleation relative to growth we can again neglect the dimerization term in **Equation A26a**. In contrast to the dimerization scenario, however, we have to keep the activation term. Transforming time via $\tau := 1 - e^{-\omega t}$ such that $\tau \in [0, 1]$ and writing $a(\tau) = a(1 - e^{-\omega t}) := A(t)$ and $b(\tau) = b(1 - e^{-\omega t}) := B(t)$ the system in **Equation A26** becomes:

$$\frac{d}{d\tau}a = 1 - \frac{2}{\omega(1-\tau)}ab \,, \tag{A35a}$$

$$\frac{d}{d\tau}b = \frac{\eta}{\omega(1-\tau)}a^2 \,, \tag{A35b}$$

with the initial condition $a(0) = b(0) = 0$. The function $g(\omega, \eta)$ transforms as

$$g(\omega, \eta) = \int_0^\infty A(t)dt = \frac{1}{\omega}\int_0^1 \frac{a(\tau)}{1-\tau}d\tau \,. \tag{A36}$$

In the following we derive the asymptotic solution for $a(\tau)$ in the limit of small $\omega$ in order to evaluate the integral in **Equation A36**. In the limit $\tau \to 1$ ($\Leftrightarrow t \to \infty$) both $a(\tau)$ and $\frac{d}{d\tau}a(\tau)$ will become small whereas $b(\tau)$ increases monotonically. The reaction term in **Equation A35a** is furthermore weighted by a factor $\frac{1}{\omega}$ which will become large if $\omega \ll 1$. We therefore postulate that for sufficiently large $\tau$ the derivative $\frac{d}{d\tau}a(\tau)$ is much smaller than the two terms on the right-hand side of **Equation A35a** and hence negligible. This assumption has to be justified a

posteriori with the obtained solution. Neglecting the derivative term $\frac{d}{d\tau}a$ in (*Equation A35a*) reduces the equation to an algebraic equation and we find

$$a = \frac{\omega(1-\tau)}{2b}. \tag{A37}$$

Using this result in *Equation A35b* we can solve for $b$ by separation of variables and subsequent integration:

$$b(\tau) = (\omega\eta)^{\frac{1}{3}} \cdot \left(\frac{3}{4}\tau - \frac{3}{8}\tau^2\right)^{\frac{1}{3}}. \tag{A38}$$

From *Equation A37* we immediately obtain $a(\tau)$:

$$a(\tau) = \frac{\omega^{\frac{2}{3}}}{\eta^{\frac{1}{3}}} \cdot \frac{1-\tau}{(6\tau - 3\tau^2)^{\frac{1}{3}}} := \frac{\omega^{\frac{2}{3}}}{\eta^{\frac{1}{3}}}h(\tau), \tag{A39}$$

where by $h(\tau)$ we denote the part of the solution that depends only on $\tau$. Hence, we find that $a$ and hence also $\frac{d}{d\tau}a$ scale like $\sim\omega^{\frac{2}{3}}$, and will thus become small if $\omega \ll 1$ and $\tau$ is large enough. Therefore the solution is consistent and justifies the approximation in which we neglected the derivative term in the limit of small $\omega$ and sufficiently large $\tau$.

Note that consistency of the solution with the approximation is a sufficient criterion for the validity of the approximation: We can solve the system for $A$ and $B$ in *Equation A35* iteratively by defining

$$\frac{d}{d\tau}a_{i-1} = 1 - \frac{2}{\omega(1-\tau)}a_i b_i,$$

$$\frac{d}{d\tau}b_i = \frac{\eta}{\omega(1-\tau)}a_i^2.$$

Assuming that for $i \to \infty$, $a_i$ and $b_i$ converge to the correct solutions $a(\tau)$ and $b(\tau)$ when starting with $a_0 = 0$, we obtain $a_1$ and $b_1$ as given by *Equation A39* and *Equation A38* and can iteratively refine the approximation. The next iteration step then reads: $\frac{d}{d\tau}a_1 = 1 - \frac{2}{\omega(1-\tau)}a_2 b_2$. As $a_1 \sim \omega^{\frac{2}{3}}$ we know that the left-hand side will be small and $a_1$ and $b_1$ solve the system if the left-hand side equals 0. Writing $a_2 = a_1 + \tilde{a}_2$ and $b_2 = b_1 + \tilde{b}_2$ this gives:

$$\frac{d}{d\tau}a_1 = 1 - \frac{2}{\omega(1-\tau)}(a_1 + \tilde{a}_2)(b_1 + \tilde{b}_2) \approx \frac{-2}{\omega(1-\tau)}(a_1 \tilde{b}_2 + b_1 \tilde{a}_2). \tag{A40}$$

From dimensional analysis it follows that the correction terms $\tilde{a}_2$ and $\tilde{b}_2$ must scale like $\tilde{a}_2 \sim \omega^{\frac{4}{3}}$ and $\tilde{b}_2 \sim \omega$ and are hence much smaller than the first order approximations $a_1$ and $b_1$. Higher order corrections will give even smaller contributions showing that if $\frac{d}{d\tau}a_1 \ll 1$, $a_1$ is indeed a very good approximation.

In the limit $\tau \to 0$, however, the expression for $a(\tau)$ in *Equation A39* diverges and consistency is violated. Hence, the obtained solution is valid only for sufficiently large $\tau$.

We fix some small $\epsilon > 0$ such that the approximation can be assumed to be sufficiently good if $\frac{d}{dt}a < \epsilon$. Furthermore, we define $\tau_\epsilon$ such that $\frac{d}{d\tau}a < \epsilon$ for all $\tau > \tau_\epsilon$. Using *Equation A39* we can write this as $\frac{d}{d\tau}h < \epsilon\eta^{\frac{1}{3}}/\omega^{\frac{2}{3}}$ for all $\tau > \tau_\epsilon$, where the left-hand side, $\frac{d}{d\tau}h$, depends only on $\tau$. Hence, by decreasing $\omega$ we can make $\tau_\epsilon$ arbitrarily small: $\lim_{\omega \to 0}\tau_\epsilon = 0$. In order to calculate $g(\omega,\eta)$ the integral in *Equation A36* can be separated in a domain where the approximation $a(\tau)$ is accurate and a domain where the correct solution $\tilde{a}(\tau)$ deviates strongly from $a(\tau)$:

$$g(\omega,\eta) = \frac{1}{\omega}\int_0^{\tau_\epsilon} \frac{\tilde{a}(\tau)}{1-\tau}d\tau + \frac{1}{\omega}\int_{\tau_\epsilon}^1 \frac{a(\tau)}{1-\tau}d\tau. \tag{A41}$$

We see from **Equation A35a** that $\frac{d}{d\tau}\tilde{a}=1$ describes an upper bound to $\tilde{a}$ showing that $\tilde{a}(\tau) \le \tau$. Therefore we can bound the contribution of the first integral as $\int_0^{\tau_\epsilon} \frac{\tilde{a}(\tau)}{1-\tau}d\tau \le \int_0^{\tau_\epsilon} \frac{\tau}{1-\tau}d\tau = \frac{1}{2}\frac{\tau_\epsilon^2}{1-\tau_\epsilon}$. Because this upper bound for the integral goes to 0 if $\omega$ and hence $\tau_\epsilon$ become small the first integral will become negligible against the second one. Asymptotically, we therefore only need to consider the second integral with the solution for $a(\tau)$ as given by **Equation A39**:

$$
\begin{aligned}
g(\omega,\eta) &= (\omega\eta)^{-\frac{1}{3}}\int_0^1 (6\tau - 3\tau^2)^{-\frac{1}{3}}d\tau = (\omega\eta)^{-\frac{1}{3}}\int_0^3 \frac{dz}{6z^{\frac{1}{3}}\sqrt{1-\frac{z}{3}}} = \\
&= \frac{3^{\frac{2}{3}}\sqrt{\pi}\,\Gamma(\frac{2}{3})}{6\,\Gamma(\frac{7}{6})}(\omega\eta)^{-\frac{1}{3}} \approx 0.8969 \cdot (\omega\eta)^{-\frac{1}{3}},
\end{aligned}
\tag{A42}
$$

where we used the substitution $\tau = 1 - \sqrt{1-z/3}$ and $\Gamma(x)$ is the (Euler) Gamma function. So, in the limit of small $\omega$, $g$ scales with $\omega$ and $\eta$ with identical exponent $-\frac{1}{3}$. This contrasts the dimerization scenario where $g$ as well as $A$ and $B$ depend only on $\eta$ and are independent of $\omega$ (cf. **Equation A32, A33 and A34**).

## Numerical analysis and the threshold values for the rate constants

In order to confirm the results of the last two paragraphs and to see how $g(\omega,\eta)$ behaves in the intermediate regime where $\omega$ and $\eta$ are of the same order of magnitude we also investigate the function $g(\omega,\eta)$ numerically. For that purpose we numerically integrate the ODE-system for $A(t)$ and $B(t)$ in **Equation A26** for different values of $\omega$ and $\eta$ with a semi-implicit method. Subsequently, we integrate the solution $A(t)$ using an adaptive recursive Simpson's rule. Plotting $g$ in dependence of $\omega$ for fixed $\eta$ on a double-logarithmic scale reveals a rather simple bipartite form of $g$, see **Appendix 3—figure 1a**:

$$g(\omega,\eta) = \begin{cases} g_1(\eta)\omega^{-\frac{1}{3}} & \omega \ll 1 \\ g_2(\eta) & \omega \gg 1. \end{cases} \tag{A43}$$

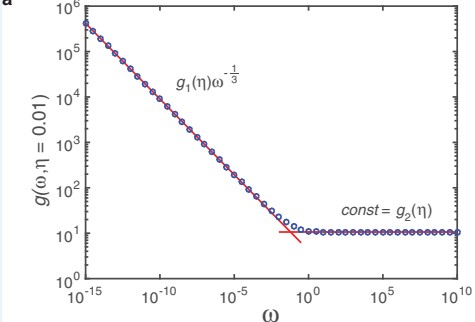 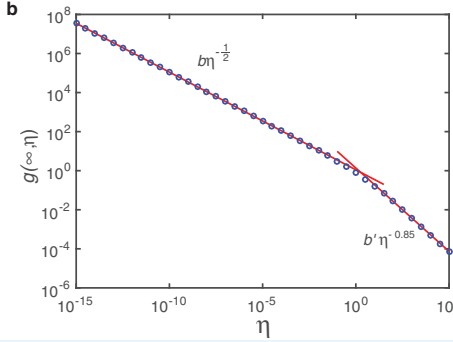

**Appendix 3—figure 1.** Fit of $g(\omega,\eta)$ on log-log scale. The function $g(\omega,\eta) = \int_0^\infty A_{\omega,\eta}(t)dt$ describes (half) the travelled distance of the profile of the polymer size distribution in dependence of $\omega = \frac{\alpha}{\nu C}$ and $\eta = \frac{\mu}{\nu}$. Marker points show solutions for $g(\omega,\eta)$ as obtained numerically from integration of **Equation A26**. Red lines are linear fits on log-log scale. In (**a**) we plot $g(\omega,\eta)$ for fixed $\eta$ (here exemplarily for $\eta = 0.01$) over 25 orders of magnitude in $\omega$ and find a markedly bipartite behavior: For small $\omega$ the dependence on $\omega$ is perfectly matched by a

power law with exponent $-\frac{1}{3}$ and $\eta$-dependent coefficient $g_1(\eta)$, whereas for large $\omega$ it is a constant $g_2(\eta)$. (**b**) Plotting $g_2(\eta) = g(\omega = \infty, \eta)$ in dependence of $\eta$ reveals again strictly bipartite behavior. Here, however, only the branch for small $\eta \leq 1$ is relevant. With the coefficient $g_1(\eta)$ that can be determined in a similar way this leads to the final form of $g(\omega, \eta)$ as given by *Equation A46*.

The transition between these two regimes is rather sharp so that $g$ is best described in a piecewise fashion

$$g(\omega, \eta) = \max\left(g_1(\eta)\omega^{-\frac{1}{3}}, g_2(\eta)\right). \tag{A44}$$

Next, we plot the coefficients $g_1(\eta)$ and $g_2(\eta)$ against $\eta$. Here we find that $g_1(\eta) = a\eta^{-\frac{1}{3}}$ with $a = \text{const} \approx 0.90$ and $g_2(\eta)$ is again bipartite with a sharp kink in between (*Appendix 3—figure 1b*):

$$g_2(\eta) = \min\left(b\eta^{-\frac{1}{2}}, b'\eta^{-0.85}\right), \tag{A45}$$

where $b \approx 1.11$ and $b' \approx 1.37$. The transition between both regimes is at $\eta \approx 1.82$. The second regime is not relevant for self-assembly since it refers to both large $\omega$ and large $\eta$, hence the travelled distance $2g$ is too small to give finite yield in this regime. Therefore, we discard the second regime and obtain as final result

$$g(\omega, \eta) = \max\left(a(\eta\omega)^{-\frac{1}{3}}, b\eta^{-\frac{1}{2}}\right), \tag{A46}$$

with $a \approx 0.90$ and $b \approx 1.11$. This confirms perfectly the exponents as well as the coefficients found in the last two paragraphs. It is, however, surprising that there is such a sharp transition between both regimes, which allows to define $g(\omega, \eta)$ in a piecewise fashion. This behavior must be the result of a series of lower oder terms in $g(\omega, \eta)$ which are unimportant in the limits $\omega \ll \eta$ and $\eta \ll \omega$ but cause the sharp transition when $\omega$ and $\eta$ are of the same order of magnitude.

Finally, we return to our original task of finding the threshold values of the activation and dimerization rate for the onset of yield. Using our result for $g(\omega, \eta)$ in *Equation A23* we find as necessary and sufficient condition to obtain finite yield in the deterministic system:

$$2\max\left(a(\eta\omega)^{-\frac{1}{3}}, b\eta^{-\frac{1}{2}}\right) \geq L - \sqrt{L}. \tag{A47}$$

Alternatively, we can state this result as two separate conditions out of which at least one must be fulfilled to obtain finite yield:

$$2a(\eta\omega)^{-\frac{1}{3}} \geq L - \sqrt{L} \Rightarrow \quad \alpha < \alpha_{\text{th}} := P_\alpha \frac{\nu}{\mu} \frac{\nu C}{(L - \sqrt{L})^3} \tag{A48}$$

$$\text{or} \quad 2b\eta^{-\frac{1}{2}} \geq L - \sqrt{L} \Rightarrow \quad \mu < \mu_{\text{th}} := P_\mu \frac{\nu}{(L - \sqrt{L})^2} \tag{A49}$$

where $P_\alpha = 8a^3 \approx 5.77$ and $P_\mu = 4b^2 \approx 4.93$. This verifies *Equation 1* in the main text.

## Appendix 4

# Impact of the implementation of sub-nucleation reactions

In the main text we focused our discussion on irreversible binding $L_{nuc} = 2$. In this section we investigate the effect of different implementations of the sub-nucleation reactions.

In general, perfect yield is trivially achieved if the complete ring is the only stable structure. However, yield can be maximal already for smaller nucleation sizes $L_{nuc}$ depending on the explicit decay rate $\delta$. In the deterministic limit without the dimerization and activation mechanisms ($\mu = \nu$, $\alpha \to \infty$) a rapid transition from zero yield to perfect yield occurs in dependence of the critical nucleation size (see *Appendix 4—figure 1*). The threshold value in this case is approximately half the ring size and is weakly affected by the decay rate $\delta$. In order to obtain finite yield for small nucleation sizes, an extremely high decay rate would be necessary. Hence, maximizing the yield solely by increasing the nucleation size is not very feasible.

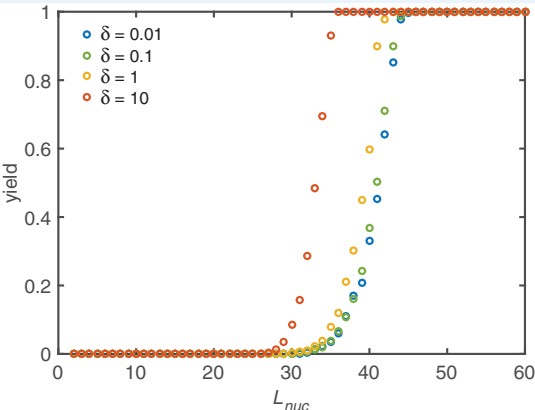

**Appendix 4—figure 1.** Yield maximization due to increased nucleation size. Without activation and dimerization mechanism ($\alpha \to \infty$, $\mu = \nu$) the yield can still be optimized by increasing the critical nucleation size $L_{nuc}$. However, a significant improvement is only achieved for critical sizes larger than half the ring size. Above, a rapid transition to perfect yield takes place. Below no effect is observed at all. Increasing $\delta$ shifts the onset of yield to slightly smaller critical nucleation sizes. Other parameters: $L = 60$, $N = 10000$.

In our model, the subcritical reaction rates $\mu_i$ may take different values. Here, we want to restrict our discussion to two scenarios. First, all rates have an identical value $\mu_i = \mu$ and second, the rates increase linearly up to the super-nucleation reaction rate:
$$\mu_i = \mu + (\nu - \mu)\frac{i-1}{L_{nuc}-1}.$$
In the deterministic limit, both implementations show the same qualitative behavior as the dimerization mechanism with $L_{nuc} = 2$ in the main text (see *Appendix 4—figure 2*). The only relevant aspect for the final yield is the extend to which nucleation is slowed down in total. In the constant scenario all reaction steps contribute equally. As a results there is a strong dependence on the number of such reaction steps, that is on the critical nucleation size. If however, the reaction rates increase linearly with the size of the polymers, the dimerization rate dominates. Only in the case $\mu \ll \nu$ finite yield is observed at all. In this limit the dimerization rate is much smaller than the subsequent growth rates. The explicit form of the different $\mu_i$ is not of major importance for the yield. The total slowdown of nucleation is the central feature. Structure decay does not play any role for intermediate nucleation sizes.

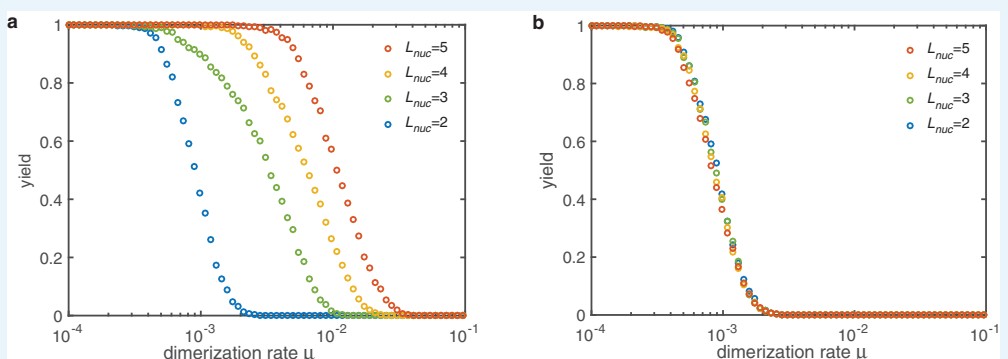

**Appendix 4—figure 2.** Yield for the dimerization mechanism ($\alpha \to \infty$) with different nucleation sizes (colors). (**a**) If all sub-nucleation growth rates are identical ($\mu_i = \mu$) increasing the nucleation size increases the threshold value $\mu_{th}$. The slow down of nucleation due to the individual sub-nucleation steps in total determines the yield. (**b**) If the sub-nucleation growth rates increase linearly $\left(\mu_i = \mu + (\nu - \mu)\frac{i-1}{L_{nuc}-1}\right)$ no dependence on the nucleation size is observed. The dimerization rate $\mu_1 = \mu$ (which is the most limiting step) dominates entirely. Other parameters: $L = 60$, $N = 10000$, $\delta = 1$.

The last question we want to address is how the combination of activation and dimerization mechanism and the corresponding non-monotonic behavior is affected by the nucleation size. Again, we compare constant sub-nucleation growth with a linearly increasing growth rate (see *Appendix 4—figure 3*). In the deterministic regime both implementations behave qualitatively similar as the dimerization mechanism discussed in the main text. However, in both cases the stochastic yield catastrophe is less pronounced. For the constant growth rates a saturation of the maximal yield is observed for sufficiently low $\mu$. If the profile is linear this effect is weaker as compared to the constant case and a dependency on the explicit value of $\mu$ is still observed. The saturation value is not reached for these reactions rates.

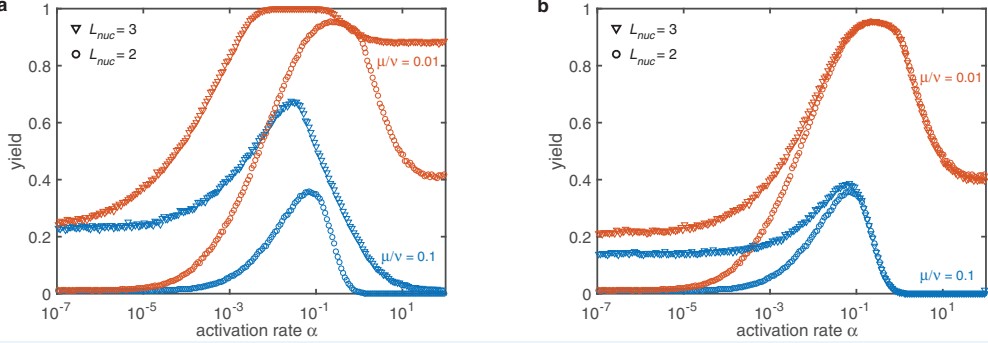

**Appendix 4—figure 3.** Combined mechanisms for different nucleation sizes (symbols) and dimerization rates (color). (**a**) If the sub-nucleation growth rates are identical ($\mu_i = \mu$) the stochastic yield catastrophe is weakened but still has a drastic impact. The qualitative behavior remains unchanged. (**b**) For a linearly increasing sub-nucleation growth rate $\left(\mu_i = \mu + (\nu - \mu)\frac{i-1}{L_{nuc}-1}\right)$ in the deterministic regime no changes are observed at all. The effect of the stochastic yield catastrophe is less pronounced. This improvement is mainly caused by structure decay which mitigates stochastic fluctuations. However, a slight dependency of the saturation value on the rate $\mu$ is observed. Other parameters: $L = 60$, $S = L$, $N = 100$, $\delta = 0.1$.

Taking all our results for the sub-nucleation behavior together we draw the following conclusions: First, structure decay by itself it not very efficient in order to maximize yield. Second, the explicit choice of the sub-nucleation rates is of minor importance for the

qualitative behavior. The system behaves similarly to the case $L_{nuc} = 2$. Third, larger nucleation sizes mitigate the stochastic yield catastrophe in general.

## Appendix 5

### Time evolution of the yield in the activation and dimerization scenario

In the main text we focus on the final yield, which represents the maximal yield that can be obtained in the assembly reaction for $t \to \infty$. Here, we briefly discuss the temporal evolution of the yield in the two scenarios. *Appendix 5—figure 1* shows the yield as a function of time for the dimerization scenario (blue) and the activation scenario (red) for the corresponding parameters indicated in the plot. Drawn lines show the evolution of the yield in the stochastic simulation whereas dashed lines represent its deterministic evolution obtained by integrating the corresponding mean-field rate equations (only shown for the activation scenario). In both scenarios, yield production sets in after a short lag time (*Hagan and Elrad, 2010*). The emergence of a lag time can be understood in terms of the interpretation of the assembly process as the progression of a travelling wave (see Sec. B). The travelling wave thereby describes the polymer size distribution and the time that is needed for the wave to reach the absorbing boundary equals the lag time for yield production observed in *Appendix 5—figure 1*. After the lag time, the yield increases very abruptly in the dimerization scenario and a bit more continually in the activation scenario. Since monomers are provided gradually in the activation scenario, the emerging wave is flatter and extends over a larger range (in polymer size space) as compared to the dimerization scenario. Consequently, yield production is more gradual in the activation scenario than in the dimerization scenario. For the same reason, the dimerization scenario is generally 'faster' or more time efficient than the activation scenario. For a detailed analysis of the time efficiency of these and other self-assembly scenarios we refer the reader to our manuscript in preparation (Gartner, Graf and Frey, in preparation).

In all depicted situations, the yield increases monotonically with time. This is, of course, generally true since the completed ring structures define an absorbing state in our system. The final yield, which is indicated in the right bar, therefore represents the upper limit for the yield that can be achieved in the assembly reaction. *Appendix 5—figure 1* shows that the temporal yield curves initially are rather steep and quickly reach a value that lies within 10% of the final yield ('quickly' thereby refers to the respective time scale), before the curves flatten and increase more slowly. This underlines that the final yield is a meaningful observable that not only describes the upper limit for the yield but also approximates the typical yield of the assembly reaction under appropriate time constraints that are not too restrictive (on the time scale set by the respective lag time).

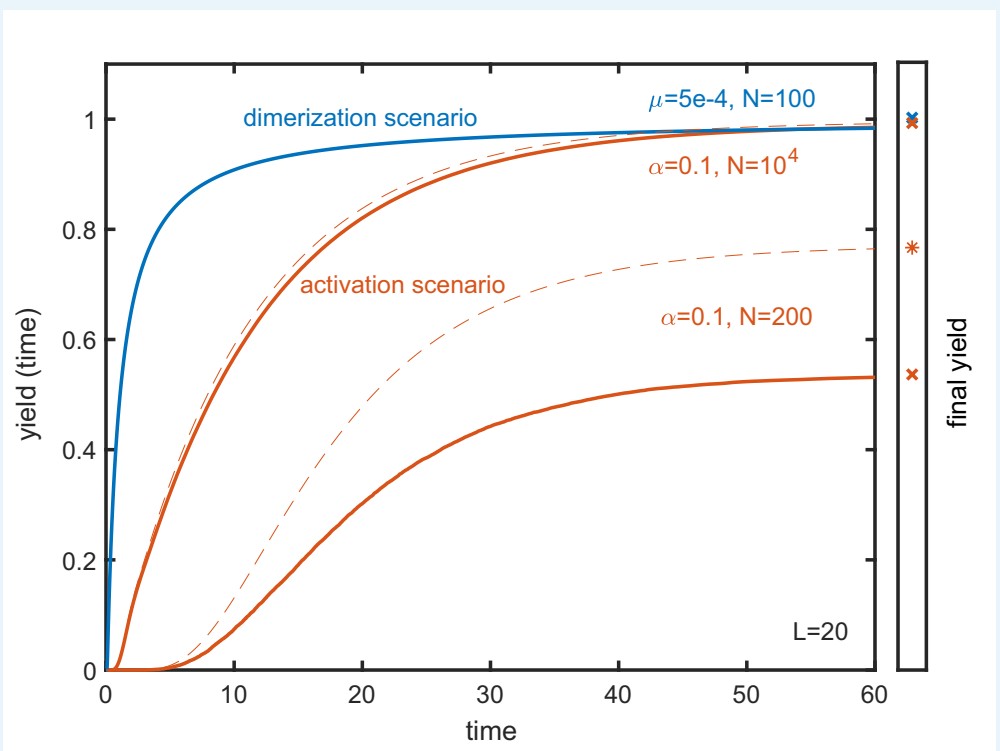

**Appendix 5—figure 1.** Time evolution of the yield in the activation and dimerization scenario. The time dependence of the yield is depicted for a dimerization scenario (blue) with $\mu = 5 \times 10^{-4}$ and $N = 100$ and for two activation scenarios (red) with $\alpha = 0.1$ and $N = 2 \times 10^2$ and $N = 10^4$, respectively, for target structures of size $L = 20$. Drawn lines show the time evolution of the stochastic systems while dashed lines describe the time evolution in the corresponding deterministic systems (where the final yield may be higher in the activation scenario). In all cases the yield increases monotonically with time. The final yield, that is indicated in the right bar, represents the upper limit of the yield at any time. Yield production in the activation scenario is generally more gradual than in the dimerization scenario. Therefore, the dimerization scenario is, in general, more time efficient than the activation scenario.

## Appendix 6

### Standard deviation of the yield

In the main text, the analysis focuses on the average yield. A priori it is, however, not apparent that this average quantity is informative, in particular due to the strong effect of stochasticity in the system. Here, we thus take a step forward to complement this picture by additionally considering a simple measure for the fluctuations of the yield, its standard deviation. *Appendix 6—figure 1* is an extension of *Figure 3a* in the main text, showing the dependence of the average yield and its sample standard deviation on the activation rate. Since yield is always positive, the standard deviation of the yield has to be small if the average yield is close to 0 ($N = 500$ in *Appendix 6—figure 1*). The same holds true for average yield close to 1 as the yield is bounded by one from above ($N = 5000$ in *Appendix 6—figure 1*). For intermediate values of the average yield, the standard deviation is highest but still small compared to the average yield ($N = 1000$ in *Appendix 6—figure 1*). The average yield is, thus, meaningful. Naturally the ratio of the standard deviation compared to the average yield also depends on the number of particles per species $N$ and on the number of species $S$. Generally speaking, for higher $N$ and $S$, this ratio decreases (see *Appendix 7—figure 1* for the dependency on $S$).

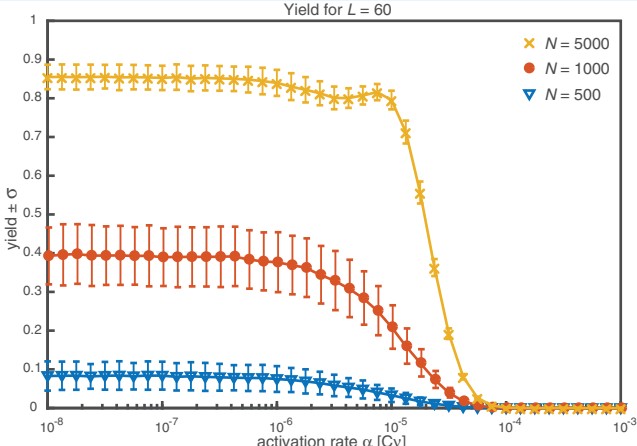

**Appendix 6—figure 1.** Average yield and its sample standard deviation. For average yield close to 0 or close to 1, the standard deviation has to be small due to the boundedness of the yield to the interval [0, 1]. For intermediate values, the standard deviation is highest. Its value is, however, still considerably smaller than the average yield. The parameters are $L = 60$, $S = L$, $\mu = \nu = 1$ and different particle numbers $N$ (colors/symbols). To obtain the average yield, the yield has been averaged over 1000 simulations. The standard deviation corresponds to the unbiased sample standard deviation.

## Appendix 7

### Influence of the heterogeneity of the target structure for fixed number of particles per species

*Figure 3d* in the main text shows how the maximal yield $y_{max}$ depends on the number of species $S$ if the ring size $L$ and the number of possible ring structures $NS/L$ is fixed. This comparison for fixed $NS$ is motivated by the question which role the heterogeneity of a structure plays for assembly efficiency if a certain number of structures should be realized. *Figure 3d* illustrates that a higher number of species $S$ (more heterogeneous structures) leads to a lower maximally possible yield, suggesting that it is beneficial to build structures with as few different species as possible. However, this situation does not correspond to the deterministically equivalent case of fixed number of particles per species $N$ (note, though, that in the deterministic case the maximally possible yield is always 1, namely for $\alpha \rightarrow 0$). Instead, for higher number of species $S$, the number of particles per species $N \propto 1/S$ decreases. How does the heterogeneity of the structures $S$ alter the maximally possible yield if $L$ and $N$ (instead of $L$ and $NS$) are fixed? *Appendix 7—figure 1* shows how the maximal yield $y_{max}$ and its standard deviation (obtained as average yield and sample standard deviation for $\alpha = 10^{-8}$ when the yield has well saturated and the dynamics (except for the timescale) get independent of the exact value of the rate-limiting activation rate) depend on the number of species $S$. For homogeneous structures $S = 1$ yield is always perfect since in this case there can be no fluctuations between species. As a result, the average yield is 1 and the standard deviation is 0. For increasing $S$, the average yield decreases until it levels off for $S \gg 1$. This behavior indicates that indeed the decreasing number of particles per species $N$ for larger $S$ is essential for the decrease of the maximal yield with $S$ in *Figure 3d*. As mentioned above, the standard deviation is largest for small $S>1$ and decreases with $S$.

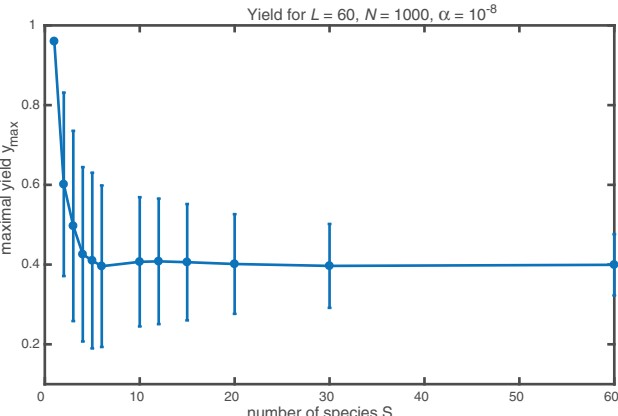

**Appendix 7—figure 1.** Influence of the heterogeneity of the target structure on the yield for fixed number of particles per species $N$. The maximal yield and its standard deviation (obtained as average yield and sample standard deviation for $\alpha = 10^{-8}$) are plotted against the number of species $S$ making up the structure of size $L = 60$. The number of particles per species $N = 1000$ is fixed. Yield drops from a perfect value of 1 for $S = 1$ to a smaller value and levels off for $S \gg 1$. The standard deviation is largest for small $S$ (except for $S = 1$ where the yield is always perfect) and decreases with increasing number of species.

## Appendix 8

### Dependence of the maximal yield $y_{\max}$ in the activation scenario on $N$ and $L$

*Figure 3c* in the main text characterizes the dependence of the maximal yield $y_{\max}$ in the activation scenario as a 'phase diagram' distinguishing different regimes of $y_{\max}$ in dependence of the particle number $N$ and target size $L$. Supplementing this figure in the main text, *Appendix 8—figure 1* shows the maximum yield that is obtained in the activation scenario in the limit $\alpha \to 0$ for fixed $L$ in dependence of $N$ (*Appendix 8—figure 1a*) as well as for fixed $N$ in dependence of $L$ (*Appendix 8—figure 1b*). For larger particle number $N$, the maximal yield exhibits a transition from 0 to 1 over roughly three orders of magnitude. Increasing $L$ shifts the transition to larger $N$. The threshold particle number where the transition starts is characterised by $N_{\mathrm{th}}^{>0}(L)$ (see main text). Approximately, for $L \le 600$, we find $N_{\mathrm{th}}^{>0}(L) \sim L^{2.8}$ (cf. main text, *Figure 3c*). Similarly, decreasing the target size $L$ for fixed $N$, the maximal yield exhibits a transition from 0 to 1 over roughly one order of magnitude in $L$. The corresponding threshold value $L_{\mathrm{th}}^{>0}$ as a function of $N$ is obtained as the inverse function of $N_{\mathrm{th}}^{>0}(L)$. Hence, at least for $N \le 10^5$, approximately it holds $L_{\mathrm{th}}^{>0}(N) \sim N^{0.36}$. Since $y_{\max}$ is largely independent of the number of species $S$ for fixed $N$ and $L$ (see Appendix 7), the maximal yield in the activation scenario (for $L_{\mathrm{nuc}} = 2$) can be fully characterized as a function $y_{\max}(N, L)$ of $N$ and $L$. Hence, $y_{\max}$ can roughly be expressed in terms of the threshold particle number $N_{\mathrm{th}}^{>0}(L)$ as

$$y_{\max}(N,L) \begin{cases} \approx 1 & \text{if } N > 10^3 N_{\mathrm{th}}^{>0}(L) \\ < 1 & \text{if } N_{\mathrm{th}}^{>0}(L) < N < 10^3 N_{\mathrm{th}}^{>0}(L) \\ = 0 & \text{if } N < N_{\mathrm{th}}^{>0}(L) \end{cases} \tag{A50}$$

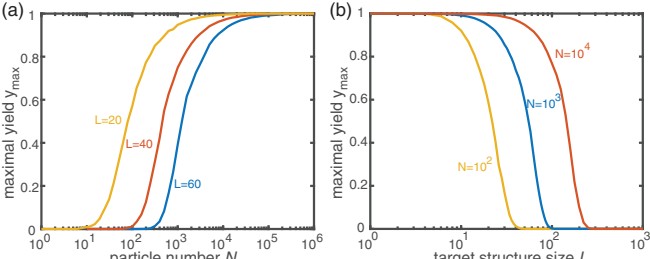

**Appendix 8—figure 1.** Dependence of the maximal yield $y_{\max}$ in the activation scenario on $N$ and $L$. For each data point, $y_{\max}$ was determined as the average yield of 100 independent stochastic simulations of the activation scenario with $\alpha = 10^{-12}$. (a) Variation of the particle number $N$ for different target sizes $L$. The maximal yield increases from 0 to 1 over roughly three order of magnitude in $N$. The onset of the transition depends on $L$. (b) Variation of the target size $L$ for different particle numbers $N$. Increasing the target size $L$ with $N$ being fixed causes the maximal yield to drop to 0. The transition from 1 to 0 spans roughly one order of magnitude in $L$ and its position is determined by $N$.

As can be seen from *Figure 3c* in the main text, the transition line between zero and nonzero yield slightly flattens with increasing $L$. Hence, the power law $N_{\mathrm{th}}^{>0}(L) \sim L^{2.8}$ (and similarly for $L_{\mathrm{th}}^{>0}$) only holds approximately and for a restricted range in $L$ and $N$. The asymptotic behavior of $N_{\mathrm{th}}^{>0}$ in the limit $L \to \infty$ remains elusive.

