## [Decision Letter]

**Acceptance summary:**

The authors study the role of fluctuations for the yield of self-assembly of heterogenous molecular complexes in specific arrangements. The approach captures the combination of subunit activation, nucleation of complex formation and complex assembly, starting from a set of different subunits in solution. This study shows in particular that when nucleation is limited by activation of subunits, yield can drop catastrophically due to stochasticity of activation and nucleation. This occurs in regimes where yield would be high when fluctuations are neglected. This work provides deep new insights in the role of stochasticity for the reliable self-assembly of molecular complexes and a framework for the study of molecular assembly in cells.

**Decision letter after peer review:**

Thank you for submitting your article "Stochastic yield catastrophes and robustness in self-assembly" for consideration by *eLife*. Your article has been reviewed by three peer reviewers, one of whom is a member of our Board of Reviewing Editors, and the evaluation has been overseen by Naama Barkai as the Senior Editor. The following individuals involved in review of your submission have agreed to reveal their identity: Pablo Sartori (Reviewer #2).

The reviewers have discussed the reviews with one another and the Reviewing Editor has drafted this decision to help you prepare a revised submission.

Summary:

The authors study the role of fluctuations for the yield of self-assembly of heterogenous molecular structures. An elegant and simple model is introduced which captures fluctuations of both activation of monomers and stochastic nucleation. The paper first recapitulates known results that slow nucleation is key to high yield. It is shown that in the deterministic or mean field limit slow nucleation by activation or by cooperative assembly (dimerization) play a similar role. However, in the presence of fluctuations, both scenarios can show very different behaviors. In particular the yield can drop in the activation scenario due to fluctuations which the authors term stochastic yield catastrophe. Interesting is the general case where activation and dimerization occur together. The authors show that in this case there can be an optimal activation rate at which yield is maximized. The work provides fundamental new insights in the self-assembly of macromolecular structures in cells.

However, the referees also raised several points that need to be addressed in a revision. In particular the depth and relevance of some of the results is not fully clear. The authors should address the specific points given below.

Essential revisions:

The relevance and strength of the results remains partly unclear. Several points need clarification:

1) The authors use the term "Stochastic yield catastrophe" to describe the drop of yield for increased fluctuations. In contrast to the catastrophic drop of yield in the deterministic case, it is unclear whether for these stochastic effects the term stochastic yield catastrophe is meaningful and appropriate as the drop is gradual and seems soft rather than representing a transition.

2) In Figure 3D the effect of increasing the number of species for a fixed size L is studied. It is shown that more species results in lower maximal yield. This could simply be an effect of the constraint *NS* = constant, which results in a decreasing number of components per species, *N*. The authors should check whether, when keeping *N* = constant, changing *S* still has an effect on the yield.

3) Still in Figure 3, the characterization of the "stochastic yield catastrophe" seems incomplete in view of its relevance to the authors. It seems that there is a certain dependence y_max_*(N,L,S)* (forgetting about *L_nuc_* for now). Immediate questions that are not addressed are: how does y_max_ scale with *N*, for fixed *L* and *S*? How does it then scale with L? And how with the heterogeneity, *L/S*? One step in this direction is Figure 3D, in which the authors show y_max_*(S)* for several values of *L*, but variable *N* (why then the label *N* = 1000 in the panel?). More such plots should be presented to clarify the key properties of the system.

4) There are additional questions which relate to the relevance of the results in realistic situations. The model is elegant but very much simplified. For example, growth is fully reversible which may not be the case in many systems of macromolecular assembly. It is unclear how general and robust the results of the simplified model are if the simplifications are relaxed.

5) The time needed for assembly is not discussed. In a biological context it is essential that assembly occurs sufficiently fast. High yield in the long term alone is not enough for assembly to happen. I understand that only looking at yield at long times is an elegant simplification of the problem, but it remains unclear how useful these insights are for real situations.

---

## [Author Response]

Essential revisions:The relevance and strength of the results remains partly unclear. Several points need clarification:1) The authors use the term "Stochastic yield catastrophe" to describe the drop of yield for increased fluctuations. In contrast to the catastrophic drop of yield in the deterministic case, it is unclear whether for these stochastic effects the term stochastic yield catastrophe is meaningful and appropriate as the drop is gradual and seems soft rather than representing a transition.

The question alerted us to the fact that we have to be more explicit about the reason why we use the term’ catastrophe’. The stochastic yield catastrophe is not the change in yield with respect to the parameter *α*. The term is intended to refer to situations in which we deterministically expect a perfect yield but due to stochastic effects we may end up with zero yield instead. This drop in yield of up to 100% compared to the deterministically expected value is what we refer to as a ’stochastic catastrophe’. We have revised the definition of the term in the subsection “Stochastic effects in the case of reduced resources” of the main text to make this point clearer. Furthermore, we extended our discussion of the nonlinearities in the stochastic effects with respect to different model parameters (especially the ring size *L*) to give a more complete picture of their significance as well. This directly links to answer 3 in which those aspects are discussed in detail.

– We have adapted the introduction of the term “Stochastic yield catastrophe” in subsection “Stochastic effects in the case of reduced resources”.

2) In Figure 3D the effect of increasing the number of species for a fixed size L is studied. It is shown that more species results in lower maximal yield. This could simply be an effect of the constraint NS = constant, which results in a decreasing number of components per species, N. The authors should check whether, when keeping N = constant, changing S still has an effect on the yield.

Indeed, if *N* (instead of *NS*) is fixed in Figure 3D, the maximal yield becomes independent of *S* for large *S >>* 1. However, this saturation value of the yield differs from the yield for the homogeneous system *S* = 1, where the maximal yield is always 1 as the system behaves deterministically. So, the yield (sharply) drops from a perfect value of 1 (for *S* = 1) to a non-perfect value for *S >* 1 and then levels off. Intriguingly, for some cases, one even observes non-monotonic behavior of the yield with respect to the number of species: Yield decreases from 1 (for *S* = 1) to a lower value and then increases again before it saturates for *S >>* 1. This rather non-intuitive behavior needs a more thorough discussion which we provide in detail in a follow-up manuscript (in preparation). In short, we believe that similar to the deterministic description also the stochastic limit shows equivalent behavior for different number of species *S*, as long as *N* and *L* are fixed and *S >>* 1.

In this manuscript, we decided to focus on the case where *NS* = *const* since this ensures that all compared systems can build the same number of rings, *NS/L*. In this context, Figure 3D suggests that in order to achieve high yield, it is beneficial to build structures that are as homogeneous as possible. So, we believe that the relevance of our findings does not change. Nonetheless, in order to pick up the important point made by the referees, we added a remark to the manuscript and a figure that shows the flattening of the maximal yield for *S >>* 1.

– We have added a remark to subsection “Stochastic effects in the case of reduced resources”.

– We have added subsection “Influence of the heterogeneity of the target structure for fixed number of particles per species” and a figure to the Appendix.

3) Still in Figure 3, the characterization of the "stochastic yield catastrophe" seems incomplete in view of its relevance to the authors. It seems that there is a certain dependence y_max_(N,L,S) (forgetting about L_nuc_ for now). Immediate questions that are not addressed are: how does y_max_ scale with N, for fixed L and S? How does it then scale with L? And how with the heterogeneity, L/S? One step in this direction is Figure 3D, in which the authors show y_max_(S) for several values of L, but variable N (why then the label N=1000 in the panel?). More such plots should be presented to clarify the key properties of the system.

We are happy to take this opportunity to clarify the dependence of *y*_max_(*N,L,S*) on *N,L* and *S*. As already explained in point (2) and shown by the added plot, *y*_max_ quickly becomes independent of *S* when *S >* 1. Hence, what remains to be described is the dependence of *y*_max_ on *N* and *L*. Phenomenologically, by increasing the particle number *N* (for fixed *L*), we encounter a threshold value *N*_th_ where the yield starts to increase sigmoidally from 0 to 1 (roughly over three orders of magnitude in *N*), quite similar to the behavior of the deterministic yield with respect to *α* and *µ* in Figure 1. Analogously, for fixed particle number *N*, by decreasing *L*, *y*_max_ exhibits a rapid transition from 0 to 1 at some threshold *L_th_*. We included two additional plots in the Appendix that show the dependences of *y*_max_ on *N* and *L*, respectively. In the main text, in Figure 3C, we characterize the dependence of *y*_max_ on *N* and *L* in combined form as a “phase diagram” that shows the four regimes *y*_max_ = 0, 0 *< y*_max_
*<* 0.5, 0.5 *< y*_max_
*<* 0.99 and 0.99 *< y*_max_ in *N*−*L* phase space (we now include the regime 0.99 *< y*_max_ to show that the transitions from 0 to almost perfect yield extend over a finite range in *N* or *L*, respectively).

In order to assess the relevance of stochastic effects for a particular system it is, therefore, crucial to know *N*_th_(*L*), the threshold particle number as a function of L, whose inverse gives *L*_th_(*N*). The function *N*_th_(*L*) is exactly described by the line that separates the two regimes *y*_max_ = 0 and *y*_max_
*>* 0 in the “phase diagram” in Figure 3C. On the double logarithmic scale, we find that there is an approximate power-law dependence of *y*_max_ on *L* with an exponent of around 2.8. It is not totally clear to us why there is this dependence with such a high exponent. Simple scaling arguments that we tried could only explain an exponent of maximally 1.5. We investigate the underlying reasons for this dependence in a technical follow-up project (manuscript in preparation). The strongly nonlinear dependence of *N*_th_ on *L* partly explains why we argue that the expression “stochastic yield catastrophe” is justified, as the relevance of this effect strongly increases with the size of the target structure. We extended the caption of Figure 3C and the description in the main text in order to clarify the relevance of the stochastic yield catastrophe and the dependence of *y*_max_ on the system parameters.

The label “*N* = 1000” in Figure 3D was, of course, incorrect and misleading. We thank the attentive referee for this remark and have corrected the label.

– We have included the regime *y*_max_
*>* 0.99 in Figure 3C and extended the description in the Figure caption and the main text.

– We have added subsection “Dependence of the maximal yield *y*_max_ in the activation scenario on *N* and *L*” and two additional figures to the Appendix.

– We have corrected the false label in Figure 3D.

4) There are additional questions which relate to the relevance of the results in realistic situations. The model is elegant but very much simplified. For example, growth is fully reversible which may not be the case in many systems of macromolecular assembly. It is unclear how general and robust the results of the simplified model are if the simplifications are relaxed.

In our model two different modes of binding exist. Below a polymer size *L*_nuc_ binding is fully reversible whereas above *L*_nuc_ irreversible growths takes place. This irreversibility is meant to represent a separation of time scales between assembly and disassembly of a stable structure. Introducing a very small dissociation rate would not affect our results significantly on the time scale of observation (see answer to question 5) but makes the definition of the yield ambiguous. If binding is reversible for all unfinished structures (*L*_nuc_ = *L* − 1), trivially, only a yield of 1 is possible at the end of the assembly process. However, the process then may take an unphysical amount of time. The relevance of the size of *L*_nuc_ for the stochastic yield catastrophe is studied in Figure 3D. The robustness of all other effects with respect to the size of *L*_nuc_ (which corresponds to the degree of reversibility) is discussed in Appendix 4. Furthermore, the two additional models illustrated in Figure 6A and Figure 7A were introduced to test for robustness against other model assumptions like monomer binding and linear growth. In all cases, we still find stochastic yield reduction, which leads us to the conclusion that the observed effects do not rely on model specifics. We added remarks to the main text to clarify the relation between *L*_nuc_ and the existence of reversible and irreversible binding. Finally, we want to remark that the assumption of a finite *L*_nuc_
*< L* and irreversible growth above this threshold have been applied successfully in experimental studies, as for example of the Brome Mosaic Virus Capsids (Chevance and Hughes, 2008).

– We made the relation between reversible and irreversible binding and *L*_nuc_ more explicit in the main text.

5) The time needed for assembly is not discussed. In a biological context it is essential that assembly occurs sufficiently fast. High yield in the long term alone is not enough for assembly to happen. I understand that only looking at yield at long times is an elegant simplification of the problem, but it remains unclear how useful these insights are for real situations.

The importance of time in the biological context is a very pertinent observation and we fully agree with the referee that this question arises naturally: time efficiency plays a significant role in biological and also artificial self-assembly. Due to its significance and due to the following three reasons we believe the question of assembly time and time efficiency is beyond the scope of the current study. First, a thorough discussion of time in the context of self-assembly turned out to be extensive and hence would have considerably increased the length of the manuscript. Second, a simultaneous discussion of both factors, stochasticity and time efficiency, would complicate the analysis (if not treated separately) and obscure the significance and relevance of each one of these factors. Third, the final yield still is a most significant observable that characterizes the efficiency of the assembly process in an explicit way: Since the yield monotonically increases with time, the final yield represents an upper bound for the yield at any finite time. In other words, the final yield is the maximum yield that can be achieved in the assembly process irrespective of time constraints. As such, the final yield is an important and informative observable that can be defined and analyzed in an unambiguous way. For those reasons, we decided to focus here entirely on the analysis of the final yield and how it is affected by stochastic effects and to treat the time aspect separately.

However, in order to give the reader an impression of what the time evolution of the yield typically looks like, we now included a corresponding plot and a new section in the Appendix. The plot exhibits typical time trajectories of the yield in both scenarios and shows, in particular, that the yield increases monotonically with time. It also shows that typically, after some delay time, the yield increases rather quickly to a value that lies within 10% of the final yield and then continues to grow more slowly. This underlines that the final yield is a meaningful observable that describes the typical outcome of the assembly reaction well, under appropriate time constraints that are not too restrictive.

– We have added a paragraph to clarify the significance of the observable of the final yield and to state more explicitly that the time efficiency is not considered here.

– We have added subsection “Time evolution of the yield in the dimerization and activation scenario” and a figure to the Appendix.